

# Influence of Pleistocene climatic oscillations on the phylogeography and demographic history of endemic vulnerable trees (section *Magnolia*) of the Tropical Montane Cloud Forest in Mexico

Yessica Rico[1,2], M. Ángel León-Tapia[3], Marisol Zurita-Solís[1],
Flor Rodríguez-Gómez[4] and Suria Gisela Vásquez-Morales[5]

[1] Red de Diversidad Biológica del Occidente Mexicano, Instituto de Ecología A.C., Pátzcuaro, Michoacán, México
[2] CONACYT, Ciudad de México, México
[3] Laboratorio de Sistemática Filogenética, Red de Biología Evolutiva, Instituto de Ecología A.C., Xalapa, Veracruz, Mexico
[4] Departamento de Ciencias Computacionales, División de Electrónica y Computación, Centro Universitario de Ciencias Exactas e Ingenierías, Universidad de Guadalajara, Guadalajara, Jalisco, Mexico
[5] Departamento de Biología, División de Ciencias Naturales y Exactas, Universidad de Guanajuato, Guanajuato, Guanajuato, México

Corresponding author
Yessica Rico, yessica.rico@inecol.mx

## ABSTRACT

The Tropical Montane Cloud Forest (TMCF) is a highly dynamic ecosystem that has undergone frequent spatial changes in response to the interglacial-glacial cycles of the Pleistocene. These climatic fluctuations between cold and warm cycles have led to species range shifts and contractions-expansions, resulting in complex patterns of genetic structure and lineage divergence in forest tree species. In this study, we sequenced four regions of the chloroplast DNA (*trnT-trnL, trnK5-matk, rpl32-trnL, trnS-trnG*) for 20 populations and 96 individuals to evaluate the phylogeography, historical demography, and paleodistributions of vulnerable endemic TMCF trees in Mexico: *Magnolia pedrazae* (north-region), *M. schiedeana* (central-region), and *M. schiedeana* population Oaxaca (south-region). Our data recovered 49 haplotypes that showed a significant phylogeographic structure in three regions: north, central, and south. Bayesian Phylogeographic and Ecological Clustering (BPEC) analysis also supported the divergence in three lineages and highlighted the role of environmental factors (temperature and precipitation) in genetic differentiation. Our historical demography analyses revealed demographic expansions predating the Last Interglacial (LIG, ~125,000 years ago), while Approximate Bayesian Computation (ABC) simulations equally supported two contrasting demographic scenarios. The BPEC and haplotype network analyses suggested that ancestral haplotypes were geographically found in central Veracruz. Our paleodistributions modeling showed evidence of range shifts and expansions-contractions from the LIG to the present, which suggested the complex evolutionary dynamics associated to the climatic oscillations of the Pleistocene.

![PeerJ logo]

Habitat management of remnant forest fragments where large and genetically diverse populations occur in the three TMCF regions analyzed would be key for the conservation of these magnolia populations.

## INTRODUCTION

Examining population genetic structure over historical spatial and temporal scales and its relationship with environmental changes is crucial for understanding species distributions and adaptations to the ongoing climatic changes (*Scoble & Lowe, 2010*; *Dalmaris et al., 2015*). This information is extremely pertinent for vulnerable ecosystems of high species diversity and endemism in tropical regions. In Mexico, the Tropical Montane Cloud Forest (TMCF) covers less than 1% of the Mexican territory but has the highest biotic diversity per unit area that nearly accounts for 10% of the flora (*Rzedowski, 1978*, *1996*) and 12% of the terrestrial vertebrates at the national level (*Pineda et al., 2005*; *Sánchez-González, Morrone & Navarro-Sigüenza, 2008*). The TMCF usually occurs between 1,200 to 2,500 m.a.s.l., and with a patchy distribution in narrow strips along mountainous ranges (*Rzedowski, 1978*; *Alcántara, Luna & Velázquez, 2002*). This ecosystem harbors highly specialized species dependent on microclimatic conditions associated to the presence of fog, high atmospheric humidity, and frequent rainfall (1,000–5,000 mm) (*Rzedowski, 1978*; *Cruz-Cárdenas et al., 2012*), which implies that the TMCF is very vulnerable to climate change (*Ponce-Reyes et al., 2012*).

The TMCF is a highly dynamic ecosystem, which has undergone frequent spatial changes in response to physical and climatic phenomena over geological time scales since the Neogene (*Graham, 1999*; *Rahbek et al., 2019*), and with acute changes during the interglacial-glacial cycles of the Quaternary (*Ramírez-Barahona & Eguiarte, 2013*; *Guevara, 2020*). During this period, the climatic variations between cold and warm cycles resulted in expansions and contractions of the TMCF due to the species dependency to the high atmospheric humidity (*Ramírez-Barahona & Eguiarte, 2013*). These processes led to complex patterns of connectivity and fragmentation on species distributions, evolutionary adaptations to local environmental conditions and lineage divergence in plant and animal populations (*Gutiérrez-Rodríguez, Ornelas & Rodríguez-Gómez, 2011*; *Ornelas et al., 2013*; *Venkatraman et al., 2019*; *Rahbek et al., 2019*). Species demographic dynamics of the TMCF in response to the climatic oscillations of the Pleistocene in the Neotropics have been explained by two main precipitation models (reviewed in *Ramírez-Barahona & Eguiarte, 2013*). The dry refugia model states that the cool conditions during the Last Glacial Maximum (LGM ~23 ka) led to downslope migrations but the arid conditions in the lowlands resulted in species range contractions into glacial refugia. Subsequently in the Holocene (~11.7–8.3 ka) with the increase in temperature and humidity, populations expanded and re-colonized their former

distribution ranges (*Prance, 1982*; *Ramírez-Barahona & Eguiarte, 2013*). According to the dry refugia and based on the distribution of endemic species and centers of endemism, *Toledo (1982)* postulated eight glacial refugia for forest species in Mexico: five adjacent to Central America and three in the drainage basin of the Gulf of Mexico in Veracruz and Oaxaca. Contrary to this model, the moist forests model states that the prevalence of humid conditions during the LGM lead to species migrations and expansions in the lowlands resulting in gene flow and wide range population connectivity. Later during the Holocene, the increase in temperature would have led to species range fragmentations and contractions into high altitude regions (*Ramírez-Barahona & Eguiarte, 2013*). Each model would have led to different genetic signatures that can be contrasted through analyses of phylogeography, demography, and ecological niche modeling (*Ramírez-Barahona & Eguiarte, 2013*; *Ornelas et al., 2019*).

The most relevant genetic signatures according to the dry refugia are the genetic differentiation of populations from separate refugia due to limited gene flow during the LGM, a star-shaped allele genealogy of post-glacial expanding lineages and thus evidence of demographic expansions. Decline of genetic diversity away from refugial regions might also be expected by the occurrence of founder effects, whereas rare alleles would occur in high frequency in refugial populations (*Excoffier, Fol & Petit, 2009*; *Gutiérrez-Rodríguez, Ornelas & Rodríguez-Gómez, 2011*; *Ramírez-Barahona & Eguiarte, 2013*). For the moist forests model, the stability of humid conditions during the LGM, which increased population connectivity, would have led to the maintenance of genetic diversity and homogenization of genetic variation with no clear geographical structuring (*Twyford et al., 2012*; *Ramírez-Barahona & Eguiarte, 2013*). Demographic expansions and bottlenecks would be less likely to occur in large and continuous populations. Genetic evidence shows mixed support for both models (see *Ramírez-Barahona & Eguiarte, 2013*) and for alternative hypotheses (see *Ornelas et al., 2019*; *Salces-Castellano et al., 2021*) in TMCF plant and animal species. Thus, gathering more data from other relevant endemic species is needed to better understand the complex dynamics on the evolution of the TMCF.

Trees are good models to evaluate demographic and phylogeographic patterns due to their long generation times that more likely preserve historical demographic signals. Moreover, they provide foundational habitat for many specialized endemic flora and fauna, being key elements of the TMCF biodiversity. Magnoliaceae is an ancient family of flowering trees and shrubs (subfamily divergence ~78–47 Mya between the Cretaceous and the Eocene; *Ramírez-Barahona, Sauquet & Magallón, 2020*) that had a continuous distribution in North America and Asia in the Eocene climatic optimum consistent with the Boreotropical flora (*Dong et al., 2021*). During the Neogene, magnolias migrated southward to warmer and wetter conditions in high valleys and mountain belts in Mexico, Central and South America (*Hebda & Irving, 2004*). In Mexico, magnolias diversified with approximately 40 recognized species occurring in humid forests, such as the TMCF and other ecosystems (*Vázquez-García et al., 2017*). Fossil records of *Magnolia* seeds suggest they were abundant and widespread in the Northern Hemisphere in the

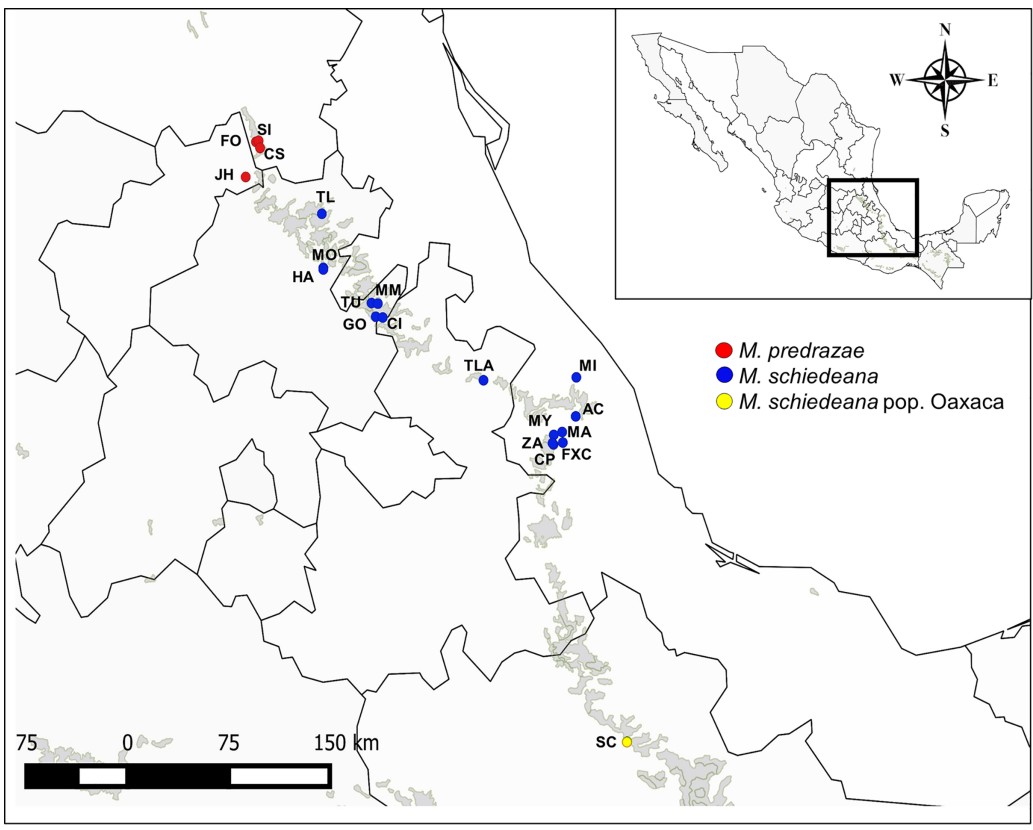

**Figure 1 Sampled localities of *M. pedrazae*, *M. schiedeana* and *M. schiedeana* pop. Oaxaca across the TMCF in Mexico.** State abbreviations see in Table 1.

Tertiary (*Azuma et al., 2001*), but currently many species are geographically restricted and with small population sizes (*Rivers et al., 2016*).

A representative magnolia species of the TMCF is *Magnolia schiedeana* Schltl., a narrow endemic from the late Miocene (~10 Mya; *Dong et al., 2021*) occurring in the largest strip of the TMCF in the Sierra Madre Oriental from Hidalgo to central Veracruz (*Vázquez-García, 1994*) (Fig. 1A). Due to its declining populations by anthropogenic threats of the TMCF, the species is listed as endangered in the IUCN The Red List of Mexican Cloud Forest Trees (*González-Espinosa et al., 2011*), as vulnerable by IUCN The Red List of Magnoliaceae (*Rivers et al., 2016*) and protected under Mexico's domestic legislation (NOM-059, *Secretaría de Medio Ambiente y Recursos Naturales (SEMARNAT), 2010*). *Magnolia schiedeana* was formerly considered to have a wider distribution, which included a northern-limit distribution in San Luis Potosí and Querétaro, and two disjoint distributions in eastern Guerrero and the Sierra de Juárez in northern Oaxaca (*Vázquez-García, 1994*). Taxonomic studies based on morphological characters showed that specimens in these regions were at least five different species (Guerrero: *M. vazquezii Jiménez-Ramírez et al., 2007*; *M. guerrerensis*; *Cruz-Durán, Vega-Flores & Jiménez-Ramírez, 2008*; Oaxaca: *M. oaxacensis*; *Vázquez-García et al., 2012*; *M. zamudioi*; *Vázquez-García et al., 2013*, and San Luis Potosí and Querétaro: *M. pedrazae*; *Vázquez-García et al.*,

2013), but for some populations in Oaxaca (Metates in Santiago Comaltepec) the taxonomic identity remains unclear (A. Vázquez-García, 2019, personal communications). Given their patchy distribution along mountainous ranges, vicariance has been hypothesized as an important mechanism in the evolution of many Mexican *Magnolia* (*Vázquez-García, 1994*). Despite that the TMCF harbors a considerable number of magnolia species that are threatened by habitat loss in Mexico (*Vázquez-García et al., 2017*; *Rivers et al., 2016*), the information about their population genetics and demographic dynamics associated to the climatic changes of the Pleistocene is almost lacking; this knowledge would shed light on the potential vulnerability of *Magnolia* trees to climate change from an historical perspective (*D'Amen, Zimmermann & Pearman, 2013*).

In this study we analyzed 20 populations and 96 individuals of *M. pedrazae* and *M. schiedeana* across their distribution in the Sierra Madre Oriental from San Luis Potosí to Veracruz, and one population in Santiago Comaltepec Oaxaca, which here we denote as *M. schiedeana* pop. Oaxaca. The distributions of these magnolia populations are within the largest strip of TMCF at the east of Mexico, which here we distinguish in three regions: north (*M. pedrazae*), central (*M. schiedeana*), and south (*M. schiedeana* pop. Oaxaca) (Fig. 1). Specifically, by sequencing four chloroplast DNA regions (*trnT-trnL, trnK5-matk, rpl32-trnL, trnS-trnG*) our aims were to: (*i*) perform a phylogeographic approach to evaluate the historical signatures of the genetic divergence between *M. pedrazae*, *M. schiedeana* and *M. schiedeana* pop. Oaxaca, and (*ii*) evaluate whether demographic changes and past species distributions were influenced by the climatic oscillations of the Pleistocene.

## MATERIALS AND METHODS

### Study species

*Magnolia schiedeana* is a perennial tree (>200 years) up to 25 m in height (*Rodríguez-Ramírez et al., 2020*). The species flowers from April to June, has sexual reproduction but can reproduce clonally by sprouting from the base of the trunk (*Vásquez-Morales et al., 2017*). Seed dispersal occurs by birds and small mammals (*Watanabe, Ikegami & Horie, 2002*) and pollination by a specialist beetle *Cyclocephala jalapensis* (*Dieringer & Espinosa, 1994*). *Magnolia pedrazae* occurs in remnant TMCF fragments in the north of Querétaro and southeast San Luis Potosí, whereas nothing is known about its biology. This species is listed as endangered by the IUCN Red List (*Rivers et al., 2016*), but not included under the Mexican domestic legislation.

### Sampling and DNA sequencing

During February to November 2019, we collected leaf tissue samples from 15 remnant *M. schiedeana* populations from the states of Hidalgo, Puebla, and Veracruz. Moreover, we collected individuals from one locality in Santiago Comaltepec in Oaxaca, where its taxonomic identity is unclear. Additionally, we included four populations from *M. pedrazae* in Querétaro and San Luis Potosí, which samples were collected in a previous study in 2017 (*Rico & Becerril, 2019*) for a total of 20 populations (Table 1, Fig. 1). We randomly collected young leaf samples from up to 10 adults or juvenile's trees and with

**Table 1 Sampling locations, population code, state, species, sample size (n), haplotypes, nucleotide, and haplotype diversity of *M. pedrazae*, *M. schiedeana* and *M. schiedeana* pop. Oaxaca.**

| Location | Code | State | Species | n | Haplotypes (no. of individuals) | Nucleotide diversity (π) | Hadplotype diversity (Hd) |
|---|---|---|---|---|---|---|---|
| Las Flores | FO | SLP | *M. pedrazae* | 3 | H18(1), H20(1), **H21**(1) | 0.00047 | 1 ± 0.27 |
| La Silleta | SI | SLP | *M. pedrazae* | 4 | H31(2), **H46**(1), **H47**(1) | 0.00093 | 0.83 ± 0.2 |
| Coronel Castillo | CS | SLP | *M. pedrazae* | 4 | H01(1), H02(1), H18(1), **H19**(1) | 0.00169 | 1 ± 0.18 |
| Joya del Hielo & Yesca | JH | QRO | *M. pedrazae* | 6 | H18(1), H20(1), **H29**(1), **H30**(1), H31(1), **H32**(1) | 0.00075 | 1 ± 0.09 |
| Tlanchinol | TL | HGO | *M. schiedeana* | 5 | H02(2), H12(1), H35(1), **H48**(1) | 0.00105 | 0.9 ± 0.16 |
| La Mojonera | MO | HGO | *M. schiedeana* | 7 | H03(2), H25(1), H26(1), H35(2), **H38**(1) | 0.00067 | 0.9 ± 0.1 |
| El Hayal | HA | HGO | *M. schiedeana* | 6 | H03(3), H26(1), **H27**(1), **H28**(1) | 0.00077 | 0.8 ± 0.17 |
| Tutotepec | TU | HGO | *M. schiedeana* | 6 | H03(4), H12(1), **H49**(1) | 0.00023 | 0.6 ± 0.2 |
| Medio Monte | MM | HGO | *M. schiedeana* | 6 | H02(1), H03(1), H15(1), H25(1), H35(1), **H37**(1) | 0.0007 | 1 ± 0.09 |
| El Gosco | GO | HGO | *M. schiedeana* | 6 | H02(2), H03(1), H15(1), **H24**(1), H25(1) | 0.00068 | 0.93 ± 0.12 |
| El Cirio | CI | HGO | *M. schiedeana* | 6 | H03(2), **H11**(2), H12(1), **H13**(1) | 0.001 | 0.87 ± 0.13 |
| Tlatlauquitepec | TLA | PUE | *M. schiedeana* | 1 | **H10**(1) | – | – |
| Misantla | MI | VER | *M. schiedeana* | 1 | **H36**(1) | – | – |
| Volcán de Acatlán | AC | VER | *M. schiedeana* | 9 | H01(1), H02(1), H03(1), **H04**(1), **H05**(1), **H06**(1), **H07**(1), **H08**(1), **H09**(1) | 0.00134 | 1 ± 0.05 |
| La Martinica | MA | VER | *M. schiedeana* | 5 | H03(1), H12(1), **H33**(1), **H34**(1), H35(1) | 0.00133 | 1 ± 0.13 |
| Mesa de la Yerba | MY | VER | *M. schiedeana* | 5 | H03(1), **H39**(1), **H40**(1), **H41**(1), **H42**(1) | 0.00154 | 1 ± 0.13 |
| Cinco Palos | CP | VER | *M. schiedeana* | 4 | **H14**(1), H15(1), **H16**(1), **H17**(1) | 0.00128 | 1 ± 0.18 |
| Reserva Ecológica | FXC | VER | *M. schiedeana* | 2 | **H22**(1), **H23**(1) | 0.00035 | 1 ± 0.5 |
| El Zapotal | ZA | VER | *M. schiedeana* | 5 | H02(1), H03(2), H12(1), H35(1) | 0.00042 | 0.9 ± 0.16 |
| Santiago Comaltepec | SC | OAX | *M. schiedeana* pop Oaxaca | 5 | **H43**(3), **H44**(1), **H45**(1) | 0.0007 | 0.7 ± 0.22 |

**Note:**
Haplotypes denoted in bold are private haplotypes to the population. State abbreviations as follows: SLP San Luis Potosí, QRO Querétaro, HGO Hidalgo, PUE Puebla, VER Veracruz, OAX Oaxaca.

a minimum separation between sampled trees of 10 m. Leaves were preserved in sealable plastic bags containing silica gel until DNA extractions were performed. GPS coordinates were recorded for each locality. Permission to conduct our fieldwork was granted by the Mexican government (SEMARNAT SGPA/DGGFS/712/1062/18).

Genomic DNA from 20 mg of dried tissue was extracted following the CTAB extraction protocol of *Doyle & Doyle (1987)*. We amplified four chloroplast (cpDNA) intergenic spacers: *trnT-trnL* (744 bp), *trnK5-matk* (789 bp), *rpl32-trnL* (624 bp) (*Azuma, Chalermglin & Nooteboom, 2011*), and *trnS-trnG* (702 bp) (*Shaw et al., 2005*). PCR reactions were carried out using the MyTaq™ DNA polymerase kit (BIOLINE, London, United Kingdom) following the conditions outlined in *Rico & Becerril (2019)*. Using the forward primers, PCR products were sequenced in MAGROGEN Inc. (Seoul, South Korea). A positive and negative control were included in each PCR plate to control for contamination. The quality of sequences was revised and edited in CHROMAS v2.6.5

(Technelysium Pty Ltd., South Brisbane, QLD, Australia). We successfully sequenced 85 individuals, which unique sequences are available in NCBI GENBANK (www.ncbi.nlm. nih.gov) (NCBI accession numbers *trnK5-matk*: MW321790–MW321798, *rpl32-trnL*: MW321799–MW321808, *trnS-trnG*: MW321809–MW321812, *trnT-trnL*: MW321813– MW321827). Additionally, we used four cpDNA sequences from eight *M. pedrazae* individuals from San Luis Potosí and Querétaro and three *M. schiedeana* from Veracruz (CP population) obtained from *Rico & Becerril (2019)* for a total of 96 individuals.

## Phylogenetic and haplotype relationships

Sequences were aligned using MUSCLE v3.8.31 algorithm (*Edgar, 2004*) with default parameters and subsequently manually adjusted in MEGA X (*Kumar et al., 2018*). To reconstruct phylogenetic relationships, we included sequences from other *Magnolia* species available from GENBANK: *Liriodendron chinese* (KU170538), *M. mexicana*, Section Talauma (MN700657), *M. ovata* Section Talauma (MT293605), *M. tripetala* Section Rhytidospermum (AY727271, DQ826283, AY727517), *M. acuminata* Section Tulipastrum (MN990595), *M. virginiana* Section Magnolia (AB553858, AB553841, AB553861, AB553852), *M. iltisiana* Section Magnolia (MK210435, MK210442, MK210448, MK210453), and one sequence obtained in this study from *M. pacifica* Section Magnolia (Locality San Sebastián del Oeste, Jalisco). These species were chosen based on their closely phylogenetic relationships and *L. chinese* as the sister group of *Magnolia* (*Azuma, Chalermglin & Nooteboom, 2011*; *Wang et al., 2020*). The sequence from *L. chinese* was used as outgroup to root the tree. The four cpDNA sequences were concatenated using SEQUENCEMATRIX (*Vaidya, Lohman & Meier, 2011*). The best substitution model fitting each marker was selected using the Akaike information criterion (AIC) in JMODELTEST v2.1.10 (*Darriba et al., 2012*). A consensus phylogenetic tree was obtained by Bayesian Inference (BI) using MRBAYES v3.2.2 (*Huelsenbeck & Ronquist, 2001*), by performing two independent runs of three cold chains, one heated chain, and specifying 20,000,000 generations with sampled trees every 1,000 generations. We assessed convergence until we reached an average standard deviation of split frequencies below 0.01. We discarded the first 25% of generated trees as burn-in, and posterior probabilities (PP) were estimated from the posterior distribution of retained trees.

Genealogic relationships were obtained by constructing a median-joining (MJ) haplotype network in POPART (*Leigh & Bryant, 2015*) for the concatenated cpDNA matrix of *M. pedrazae* (n = 17), *M. schiedeana* (n = 74) and *M. schiedeana* pop. Oaxaca (n = 5). All subsequent analyses were carried for this 96 individual's data set.

## Genetic diversity, differentiation, and phylogeographical structure

For each sampled locality and for the three lineages, we estimated the mean haplotype (*Hd*) and nucleotide diversity ($\pi$) using DNASP v6.12.03 (*Rozas et al., 2017*). Analyses of molecular variance (AMOVA) were performed to evaluate the amount of genetic variance and genetic differentiation ($F_{ST}$) found in the following groupings: (1) no predefined groups, (2) populations in two lineages, *M. pedrazae* + *M. schiedeana* and *M. schiedeana* pop. Oaxaca, and (3) populations in three lineages, *M. pedrazae*, *M. schiedeana* and

*M. schiedeana* pop. Oaxaca. AMOVAs were performed using the Tamura and Nei genetic distance and 1,000 permutations to determine the statistical significance of each partition as implemented in ARLEQUIN v3.5.2.2 (*Excoffier & Lischer, 2010*). Evidence of phylogeographic structure was determined by contrasting the coefficients of population differentiation $G_{ST}$ and $N_{ST}$. A significantly higher $N_{ST}$ than $G_{ST}$ is evidence of a significant phylogeographical structure resulting from the occurrence of closely related haplotypes in populations (*Pons & Petit, 1996*). These coefficients were obtained using PERMUT v2.0 (*Pons & Petit, 1996*) with 10,000 permutations.

We used BAPS v5.3 (*Corander, Sirén & Arjas, 2008*) to determine the most likely number of genetic clusters using the clustering of linked loci module and the codon linkage model, appropriate for sequence data. First, with few replicates ($n = 5$) and three independent runs, we explored the likely number of genetic clusters from $K = 2$ to 10. By looking at the posterior probabilities and likelihood values, we observed that the most likely number of clusters was between 3 and 5 and thus the final analysis was carried out with two independent runs for $K = 2$ to 5 with 10 replicates each. The most likely number of $K$ clusters was determined by its higher posterior probability and likelihood value. Additionally, pairwise $F_{ST}$ comparisons were calculated in ARLEQUIN v3.5.2.2 with 1,000 permutations to test for statistically significant differences between lineages. Mantel correlations to test the effect of isolation by geographical distance (IBD) on $G_{ST}$ genetic and Euclidean distances was examined for the overall data set. Significance of the Mantel correlation was tested by permuting observations 1,000 times using the R library vegan 2.5.7 (*Oksanen et al., 2015*; *R Core Team, 2020*).

## Environmental differentiation on phylogeographic genetic clusters

We implemented a Bayesian Phylogeographic and Ecological Clustering analysis (BPEC) in the BPEC v1.3.1 R package to reveal the geographical distribution of genetic clusters by considering genetic, geographical, and environmental data (*Manolopoulou et al., 2011*; *Manolopoulou & Emerson, 2012*). This analysis assumes that population substructure is the result of migration events into new sites, which can be explained by geographical and ecological restrictions to gene flow (*Manolopoulou, Hille & Emerson, 2020*). The test also provides measures of uncertainties for haplotype relationships and identify likely ancestral locations (*Manolopoulou, Hille & Emerson, 2020*). We analyzed all cpDNA haplotypes, geographical locations, and current bioclimatic data extracted for each individual occurrence for 19 bioclimatic layers in 30 arc-seconds (*Karger et al., 2017*) downloaded from CHELSA (https://chelsa-climate.org).We discarded correlated bioclimatic variables ($r \geq 0.8$, see ecological niche modeling) and use the first two synthetic axes from Principal Components Analysis (PCA) as covariates. After several initial short runs, the final analysis was carried out with two maximum number of migrations and relaxation of the parsimony criterion not allowed (zero) to reach convergence. MCMC chains were run for 10 million steps with 10,000 posterior samples saved.

Additionally, statistical differences in the environmental space occupied for each lineage was tested on the two PC axes with a Multivariate Analysis of Variance (MANOVA) using

the Pillai's trace as the test statistic (*Nakazato, Warren & Moyle, 2010*). Moreover, we performed *post hoc* Tukey-HSD pairwise comparisons on each of the PC axes (*Di Febbraro et al., 2017*) using the base packages in R.

## Historical demography

To infer historical demographic changes, we used different methods. First, we estimated neutrality tests, Tajima's D (*Tajima, 1989*), Fu's F (*Fu, 1997*), and *R2* (*Ramos-Onsins & Rozas, 2002*) with 1,000 permutations using PEGAS R package (*Paradis, 2010*). Significant negative values of Tajima's D and Fu's F and positive *R2* values suggest rejection from neutrality and can be interpreted as population expansion. Second, we computed pairwise nucleotide mismatch distributions to contrast observed and expected distributions under a demographic growth-decline model using DNASP v6.12.03 (*Rozas et al., 2017*). A unimodal pairwise distribution is expected under a demographic expansion model, whereas a multimodal distribution would be expected for populations at demographic equilibrium (*Harpending et al., 1998*). Mismatch distributions were tested with the sum of square deviations (SSD) and the Harpending's raggedness index (Hri) by implementing 1,000 permutations in ARLEQUIN v3.5.2.2 (*Excoffier & Lischer, 2010*). Significant SSD and Hri values ($P \leq 0.05$) indicate deviations from the sudden expansion model. Third, we used Bayesian skyline plots in BEAST v2.4.2 (*Drummond & Bouckaert, 2015*) to assess effective population size ($Ne$) changes across time. Two independent analyses were run, one for *M. schiedeana* ($n = 74$) and one for *M. pedrazae* ($n = 17$). We used the substitution model GTR+I as selected using AIC in JMODELTEST v2.1.10 (*Darriba et al., 2012*). Other parameters used were empirical base frequencies, a relaxed clock lognormal model, one run of 20 million generations, and trees and parameters sampled every 2,000 iterations. The time axis was scaled with the substitution rates $1.59 \times 10^{-9}$ for chloroplast-wide, synonymous substitution rates described for most angiosperms (*Wolfe, Li & Sharp, 1987*). After the analysis, we viewed the log file in TRACER v1.7.1 (*Rambaut et al., 2018*) to ensure that effective sample sizes (ESS) for all priors were >200 (*Drummond & Bouckaert, 2015*). The mismatch distribution plot and the Bayesian Skyline for *M. schiedeana* pop. Oaxaca was not possible due to small sample size.

Lastly, we used an Approximate Bayesian Computation (ABC) framework implemented in DIYABC v2.1 (*Cornuet et al., 2008, 2014*) to test for distinct competing demographic scenarios. According to the results of BAPS and BPEC, we contrasted five likely scenarios. At generation $t_0$ (present), all scenarios had the three genetic lineages. The first scenario predicts that the three lineages diverge at the same time from a common ancestor ($Na$) at time $t_2$ and remained without demographic changes until time $t_0$ (stable model). Scenarios 2, 3, and 4, predict divergence at time $t_1$ between two of the lineages to subsequently merge the three lineages at time $t_2$ (divergence model). The difference between these three scenarios is the lineage diverging at time $t_2$, and subsequently the pair of lineages diverging at time $t_1$ (see results). Lastly, the fifth scenario predicts a split between *M. schiedeana* and *M. schiedeana* pop. Oaxaca at time $t_2$, then both lineages merge with *M. pedrazae* at time $t_1$ (admixture model). We considered these five scenarios as the

most plausible given the observed relationships of divergence and admixture among the three lineages. Prior to the final run, we performed six runs to compare different parameter conditions. The run that we present in the results (see below) was the run with best adjusted parameters (Supplemental Material). Each run was implemented with 2 million coalescent-based simulated datasets and different summary statistics.

For the final run and given the absence of information on the species population sizes, we used default priors. We generated three million coalescent-based simulated datasets for each of the five evolutionary scenarios considering the HKY model, and $Na$, $N1$, $N2$ and $N3$ with a uniform prior distribution with a minimum of 10 and a maximum of 100,000 values for effective population sizes; the prior distribution of the timing of events was for $t_1$ and $t_2$ 100–100,000 generations respectively and considering a 10-year generation time (*Vásquez-Morales & Sánchez-Velásquez, 2011*). The set condition ($t_2 \geq t_1$) on the prior time distribution was added up to avoid incongruences in the simulated genealogies. For the final run, we used the following summary statistics: (1) For the three groups: Number of segregating sites, mean pairwise differences, and mean number of the rarest nucleotide at segregation sites; (2) For the 1–2, 1–3 and 2–3 groups: Number of segregating sites and $F_{ST}$. We used a prior distribution of mean mutation rates of $1.59 \times 10^{-9}$ for the chloroplast (*Wolfe, Li & Sharp, 1987*). We used the default parameters for the mutation model section. Scenario posterior probabilities were evaluated using a logistic regression on the 1% of simulated datasets nearest to the observed data (*Fontaine et al., 2013*). For the best-supported scenario, we performed a model checking procedure by applying a PCA on test statistic vectors to visualize the fit between the observed and simulated datasets. Confidence of the scenario choice was assessed by simulating 500 pseudo-observed datasets (PODs) under each scenario to calculate Type I and II error rates. Finally, for the best-supported scenario, we obtained point estimates for demographic and temporal parameters using local linear regression on the 1% of simulations nearest to the observed dataset (*Cornuet et al., 2008, 2014*).

## Present and past distribution ecological niche modeling

Due to the taxonomic uncertainty for *Magnolia* species, we only used the occurrences from specimens of *M. schiedeana*, *M. pedrazae*, and samples from Oaxaca collected in this study (Table S1). We used these occurrences because they have the quality and spatial accuracy to produce models with ecological plausibility (*Galante et al., 2018*). To avoid sampling biases, the occurrences were spatially thinned at three kilometers as a larger radius would have significantly reduce the final number of occurrences; this was carried in the SPTHIN v0.1.0.1 R package (*Aiello-Lammens et al., 2015*). Additionally, a second data set was built with occurrences downloaded from the Global Biodiversity Information Facility (GBIF; http://www.gbif.org/), but due to the likely uncertainty of some of these records, we also included localities reported in previous studies, such as herbaria specimens of *M. schiedeana* (*Vásquez-Morales et al., 2014, 2017*). Specimens with insufficient locality information were discarded; all occurrences were thinned at five kilometers due to a higher density of occurrences than the training data set. The first data set included 26 occurrences used for building the niche model and the second data set

included 50 occurrences used for testing the models. As variable predictors, we used the 19 bioclimatic layers in 30 arc-seconds from CHELSA because these layers have a better performance than other layers commonly used and can increase the accuracy of species range predictions (*Karger et al., 2017*). The environmental layers were delimited to several polygons of the level II ecoregions of North America downloaded from the United States Environmental Protection Agency (https://www.epa.gov/); the ecoregions were: Mexican high plateau, eastern Sierra Madre Oriental, west humid coastal plains and hills of the Gulf of Mexico, southern Sierra Madre del Sur, east of the Mexican Plateau, east of the Trans Mexican Volcanic Belt, and south of the North American deserts. These areas were chosen as likely accessible areas.

Values from the 19 current bioclimatic layers were extracted from individual occurrences and Pearson's correlations threshold of 0.8 and 0.7 were performed with the occurrences sites to minimize variable redundancy using NTBOX v0.1.4.5 R package (*Osorio-Olvera et al., 2020*). The final uncorrelated variables according to the 0.8 Pearson's threshold were: annual mean temperature (BIO1), isothermality (BIO3), temperature seasonality (BIO4), temperature annual range (BIO7), annual precipitation (BIO12), precipitation of the driest month (BIO14), precipitation seasonality (BIO15), and precipitation of the warmest quarter (BIO18). For the 0.7 threshold, BIO4 was the only variable discarded. To have a better balance between the number of occurrences and the number of bioclimatic variables, the niche model was built with the 0.7 correlation threshold (*Dormann et al., 2012*) under the maximum entropy algorithm (*Phillips et al., 2017*) in MAXENT v3.4.1 using the KUENM v1.1.1 R package (*Cobos et al., 2019*). Some levels of model complexity were evaluated by varying the regularization multiplier (RM) from 0.5 to 10 every 0.5, and feature classes linear (L), quadratic (Q), product (P), and threshold (T) in four fixed combinations: L, LQ, LQP, and LQPT, which resulted in 80 candidate niche models. The hinge (H) feature was not considered to simplify the niche model for subsequent transferability. The evaluation was made using 10,000 background points, five percent of training data omission rate (OR), 20% for bootstrapping to calculate the partial Receiver Operating Characteristic (pROC) with 10,000 iterations. The best model was selected considering the statistically significant models, and the lowest values of corrected Akaike information criterion (AICc) and inspecting the OR. The average niche model with the best parameters was built and projected to the same area using the mean of ten replicates of bootstrap and the logistic output format.

To construct the paleodistributions, the 19 variables used for the current model were downloaded, and the features of the best current niche model were projected to the same area into past four scenarios based on the Community Climate System Model simulations. The scenarios assessed were: mid-Holocene (MH; ~8.3–4,000 years ago (ya)) at 30 s downloaded from WORLDCLIM (http://www.worldclim.com), early-Holocene (EH; ~11.7–8,000 ya) at 2.5 min resolution (*Fordham et al., 2017*) downloaded from PALEOCLIM (http://www.paleoclim.org/), Last Glacial Maximum (LGM; ~23–14,000 ya) at 30 s downloaded from CHELSA (*Karger et al., 2018*), and Last Interglacial (LIG;

~120–140,000 ya) at 30 s (*Otto-Bliesner, 2006*) downloaded from WORLDCLIM. The resolution of EH was downscaled to 30 s. The extent and dimension of all scenarios were matched with those of the current model to make further comparisons and avoid alterations in predictions (*Randin et al., 2009*). The layer sources were chosen to have the original 30 sec resolution for most scenarios.

To evaluate the risk of extrapolating to non-analogous conditions (novel environments) in each past scenario, we compare the calibration area (*i.e.*, reference data) with the four past scenarios (*i.e.*, projection data) by quantifying the extrapolation due to covariate range (NT1) and correlation change (NT2) with the ExDet method (*Mesgaran, Cousens & Webber, 2014*) implemented in NTBOX v0.1.4.5. Transferability was carried out using three methods of extrapolation: free extrapolation, extrapolation and clamping, and no extrapolation. The niche overlap was calculated using the Schoener's D metric in the geographic space (*Warren, Glor & Turelli, 2008*), which D values range from 0 (no overlap) to 1 (identical predictions). Finally, we calculated Pearson correlations between model predictions. All procedures were performed in the R libraries raster v3.3-13 (*Hijmans, 2020*) and ENMeval v0.3.0 (*Muscarella et al., 2014*).

# RESULTS

## Phylogenetic and haplotype relationships

The phylogenetic cpDNA concatenated matrix including sequences from other *Magnolia* species used as outgroup consisted of 106 individuals with a total length without gaps and missing data of 2,803 bp. The aligned matrix consisted of 147 polymorphic and 35 informative sites. The best-fitting model of sequence evolution was GTR + I. The BI tree showed that *M. iltisiana*, *M. pacifica*, *M. pedrazae*, *M. schiedeana* and *M. schiedeana* pop. Oaxaca formed a polytomy with a low support (PP = 0.5). Samples from *M. pedrazae* and *M. schiedeana* were not monophyletic, but samples from *M. schiedeana* pop. Oaxaca were grouped within well-supported subclade (PP = 1) (Fig. 2A).

For the haplotype network, the concatenated cpDNA matrix had a total length of 2,858 bp excluding gaps and missing data for 96 individuals. There were 47 polymorphic sites and 18 informative sites. A total of 49 haplotypes were identified (Table 1). Haplotype relationships revealed only two shared haplotypes (H01 and H02) between *M. pedrazae* (north) and *M. schiedeana* (central), and none with *M. schiedeana* pop. Oaxaca (south). Overall, few haplotypes were shared among populations, but most of them diverged by one mutational change (max 5 mutation steps between H02 and H13 from *M. schiedeana*). The most frequent haplotypes (H02 and H03) were present in 7 out of 15 *M. schiedeana* populations, and present only (H02) in one population from *M. pedrazae* in San Luis Potosí. Haplotype connections between *M. pedrazae* and *M. schiedeana* occurred through haplotypes from Veracruz populations (MI (H36), FXC (H22 and H23)), while the connection of the three *M. schiedeana* pop. Oaxaca haplotypes were through a missing haplotype that tied a haplotype from *M. schiedeana* in Hidalgo (GO) and *M. pedrazae* from San Luis Potosí and Querétaro (CS, JH) (Fig. 2B).

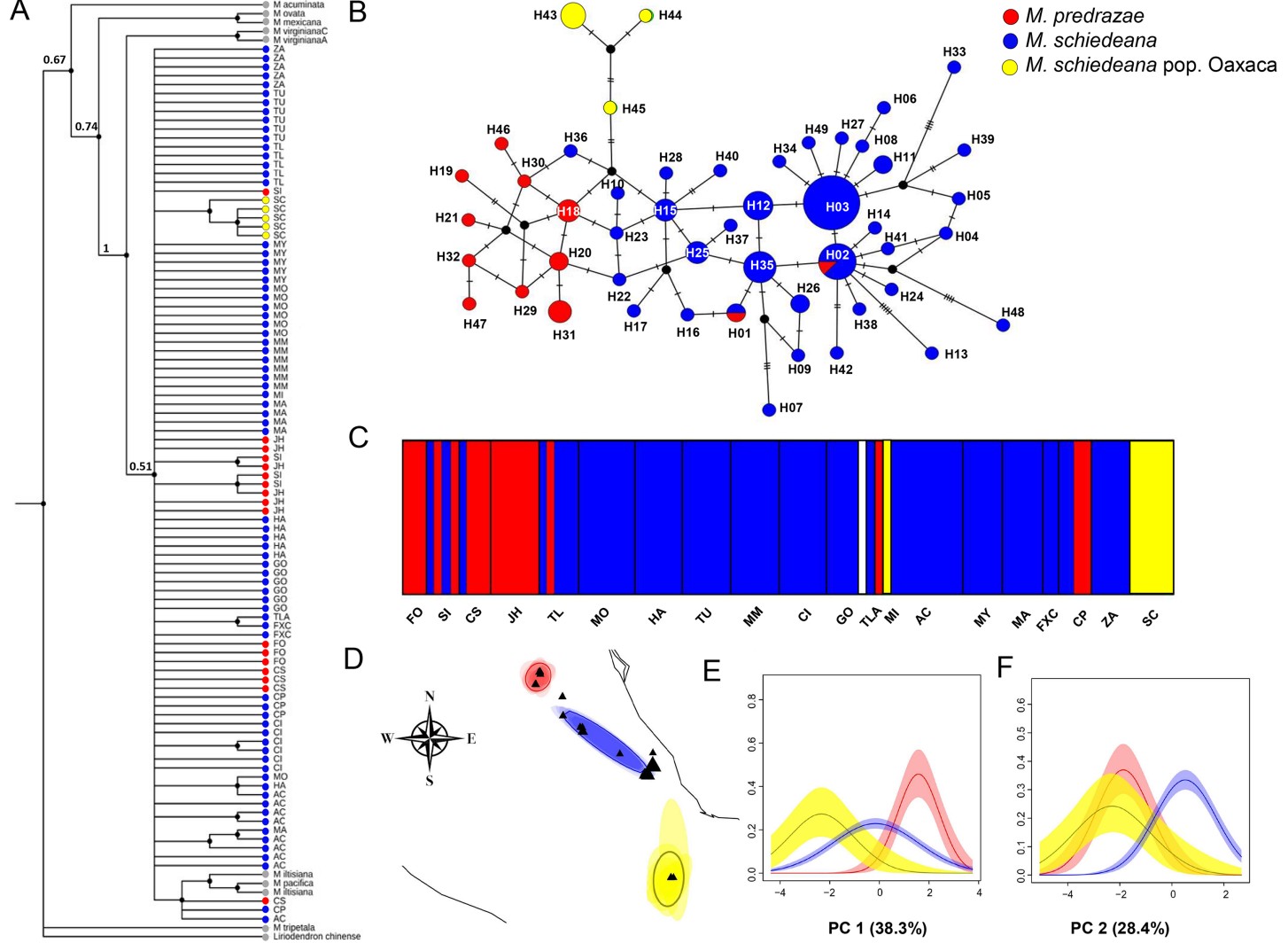

**Figure 2 Phylogenetic and haplotype relationships among *M. pedrazae*, *M. schiedeana* and *M. schiedeana* pop. Oaxaca using four concatenated cpDNA regions.** (A) Bayesian phylogenetic tree, (B) Median-joining haplotype network, (C) BAPS $K = 5$ genetic clusters, and (D) BEPC K = 3 genetic clusters and (E and F) PCA plots for eight bioclimatic variables showing the environmental differences for the three clusters. State abbreviations see in Table 1.

## Genetic diversity, phylogeographic structure and environmental differentiation

Genetic diversity was high for *M. schiedeana* (Hd = 0.923; π = 0.001) and *M. pedrazae* (Hd = 0.95; π = 0.0009) and moderate for *M. schiedeana* pop. Oaxaca (Hd = 0.7; π = 0.0007). At the level of populations, the most genetically diverse were MY and AC in Veracruz for *M. schiedeana*, and CS in San Luis Potosí for *M. pedrazae* (Table 1). The AMOVA with no groups defined *a priori* showed that the genetic variance explained among populations was 32.5% ($F_{ST}$ = 0.33, $P < 0.001$). When populations were grouped in two lineages (*M. pedrazae* + *M. schiedeana*, and *M. schiedeana* pop. Oaxaca), a much higher proportion of the genetic variance and thus genetic differentiation was observed (57.6%, $F_{CT}$ = 0.58, $P < 0.001$), whereas when three lineages were considered (*M. pedrazae*,

**Table 2 AMOVA results of distinct hierarchical groups for 20 populations of *M. pedrazae*, *M. schiedeana*, and *M. schiedeana* pop. Oaxaca.**

| Grouping | df | Sum of squares | Estimated variance | % | Fixation indices |
|---|---|---|---|---|---|
| **1. No groups defined** | | | | | |
| Among populations | 17 | 78.675 | 0.6314 | 32.46 | |
| Within populations | 77 | 101.147 | 1.3136 | 67.54 | $F_{ST} = 0.325$*** |
| Total | 94 | 179.82 | 1.945 | | |
| **2. Two lineages** | | | | | |
| Among groups | 1 | 25.126 | 2.3079 | 57.58 | $F_{CT} = 0.576$*** |
| Among populations within groups | 16 | 53.549 | 0.3864 | 9.64 | $F_{SC} = 0.227$*** |
| Within populations | 77 | 101.147 | 1.3136 | 32.78 | $F_{ST} = 0.672$*** |
| Total | 94 | 179.823 | 4.008 | | |
| **3. Three lineages** | | | | | |
| Among groups | 2 | 50.706 | 1.3217 | 48.25 | $F_{CT} = 0.483$*** |
| Among populations within groups | 15 | 27.97 | 0.1041 | 3.8 | $F_{SC} = 0.073$** |
| Within populations | 77 | 101.147 | 1.3136 | 47.95 | $F_{ST} = 0.5205$*** |
| Total | 94 | 179.823 | 2.739 | | |

Notes:
** $P < 0.01$.
*** $P < 0.001$.
Analyzed groups: (1) No predefined groups, (2) populations grouped in two lineages *M. schiedeana* + *M. pedrazae* and *M. schiedeana* pop. Oaxaca and (3) populations grouped by the three lineages corresponding *M. schiedeana*, *M. pedrazae* and *M. schiedeana* pop. Oaxaca.

*M. schiedeana* and *M. schiedeana* pop. Oaxaca), a slightly lower proportion of the genetic variance and genetic differentiation were observed relative to the two lineages grouping (48.3%, $F_{CT} = 0.48$, $P < 0.001$) (Table 2). Pairwise $F_{ST}$ comparisons showed that the highest differentiation was between *M. schiedeana* and *M. schiedeana* pop. Oaxaca ($F_{ST} = 0.67$, $P = 0.001$), and the lowest between *M. schiedeana* and *M. pedrazae* ($F_{ST} = 0.42$, $P = 0.001$) (Table S2). We found a weak and significant pattern of IBD for the whole data set ($r = 0.31$, $P = 0.022$), but which significant effect disappeared when *M. schiedeana* pop. Oaxaca was excluded from the analysis ($r = 0.015$, $P = 0.46$). Results from PERMUT revealed a significant phylogeographical structure ($G_{ST} = 0.048$ and $N_{ST} = 0.364$, $P < 0.05$).

BAPS revealed four likely clusters ($K = 4$, log marginal likelihood = −201.9484, PP = 0.72): cluster one was mostly formed by *M. pedrazae*, cluster 2 mostly with *M. schiedeana*, cluster 3 with only one individual from GO, and cluster 4 with *M. schiedeana* pop. Oaxaca and MI population; these observed clusters thus corresponded to the three TMCF regions (Fig. 2C). Similarly, BPEC showed three phylogeographic clusters with high membership posterior probabilities for most haplotypes (PP = 0.80–0.99). The three main clusters also corresponded to *M. pedrazae*, *M. schiedeana* and *M. schiedeana* pop. Oaxaca (Fig. 2D). However, uncertainty of cluster location was evident for the *M. schiedeana* pop. Oaxaca, as some *M. schiedeana* haplotypes (H10, H17, H22, H23, H36, H40) from Veracruz (except H10 from Puebla) were assigned to *M. schiedeana* pop. Oaxaca (Fig. S1). Ancestral locations were in central Veracruz (Fig. S1). The environmental variation of the two PCs revealed statistically significant differences among the three lineages (Pillai's trace = 0.78, $F_{2,93} = 30.21$, $P < 0.001$). Tukey-HSD

**Table 3 Genetic diversity, neutrality tests and mismatch distributions for *M. pedrazae*, *M. schiedeana* and *M. schiedeana* pop. Oaxaca.**

| Lineages | N | $N_H$ | π | $H_d$ | $D_T$ | Fs | R2 | SSD | Hri |
|---|---|---|---|---|---|---|---|---|---|
| *M. pedrazae* | 17 | 12 | 0.00098 | 0.949 | −0.524 | −0.386 | 0.117 | 0.0036 | 0.0258 |
| *M. schiedeana* | 74 | 36 | 0.001 | 0.923 | −2.09* | −4.621** | 0.0356** | 0.0009 | 0.0236 |
| *M. schiedeana* pop. Oaxaca | 5 | 3 | 0.0007 | 0.7 | −1.12 | −1.124 | 0.253 | 0.0939 | 0.290 |

**Notes:**
* $P < 0.05$.
** $P < 0.01$.

N, number of individuals; $N_H$, number of haplotypes; π, nucleotide diversity; $H_d$, haplotypic diversity; $D_T$, Tajima's D; $F_S$, Fu's Fs; R2, Ramos-Onsins and Rozas; SDD, differences in the sum of squares; Hri, Harpending's raggedness index.

revealed statistically significant pairwise differences between the three lineages along the PC 1 (38.3% $P < 0.001$); variable contribution was: temperature annual range (BIO7) = 27.3%, precipitation seasonality (BIO15) = 24.2%, temperature seasonality (BIO4) = 21.3% and precipitation of the driest month (BIO14) = 20.3%. For the PC 2 (28.4%) statistically significant differences were observed between *M. schiedeana* and *M. pedrazae* ($P < 0.001$) and between *M. schiedeana* and *M. schiedeana* pop. Oaxaca ($P < 0.001$); variable contribution was: annual precipitation (BIO12) = 34.6%, precipitation of the warmest quarter (BIO18) = 34.2%, and precipitation of the driest month (BIO14) = 13.2% (Figs. 2E and 2F).

## Historical demographic changes

Tajima's D and Fu's F showed statistically significant negative values, and positive significant R2 values for *M. schiedeana*, indicative of population expansion, while for *M. pedrazae* and *M. schiedeana* pop. Oaxaca, neutrality tests were non-significant (Table 3). Mismatch distributions for the observed and expected values for *M. pedrazae* and *M. schiedeana*, fit with the expectation of recent demographic expansions (Figs. 3A and 3B). These mismatch distributions were not rejected by the SSD and Hri tests (Table 3). Bayesian skyline plots of $N_e$ through time showed an increase in effective population size over time for *M. pedrazae* at ~200,000 years ago (Fig. 3C), and for *M. schiedeana* between ~300,000 to 250,000 years ago (Fig. 3D), both predating the LGM.

The ABC simulations provided the best support for scenario 2 and scenario 5 (PP: 0.35, 95% CI [0.342–0.366] and PP: 0.36, 95% CI [0.347–0.373], respectively) that performed better than the other three scenarios. Confidence estimates for scenario choice indicated that Type I errors for the best-supported scenarios were high (scenario 2: 0.62 and scenario 5: 0.64) but Type II errors were higher for scenario 5 (0.73) relative to the scenario 2 (0.51). As statistically there is no way to discriminate which of the two is the best scenario when both are equally likely (*Bertorelle, Benazzo & Mona, 2010*), we considered scenarios 2 and 5 as the most likely, although biologically they are contrasting (Fig. 4). Assuming a 10-year generation time, under scenario 2 and 5, the posterior mean parameter estimates indicated that the divergence ($t_2$) of *M. schiedeana* pop. Oaxaca from *M. schiedeana* occurred 82,300 (CI [37.7–99.2]) and 84,800 (CI [41.7–99.4]) years ago, respectively, which fits the LIG. Under scenario 2, the mean estimated divergence time of *M. schiedeana* and *M. pedrazae* was 35,000 (CI [3.5–81.3])

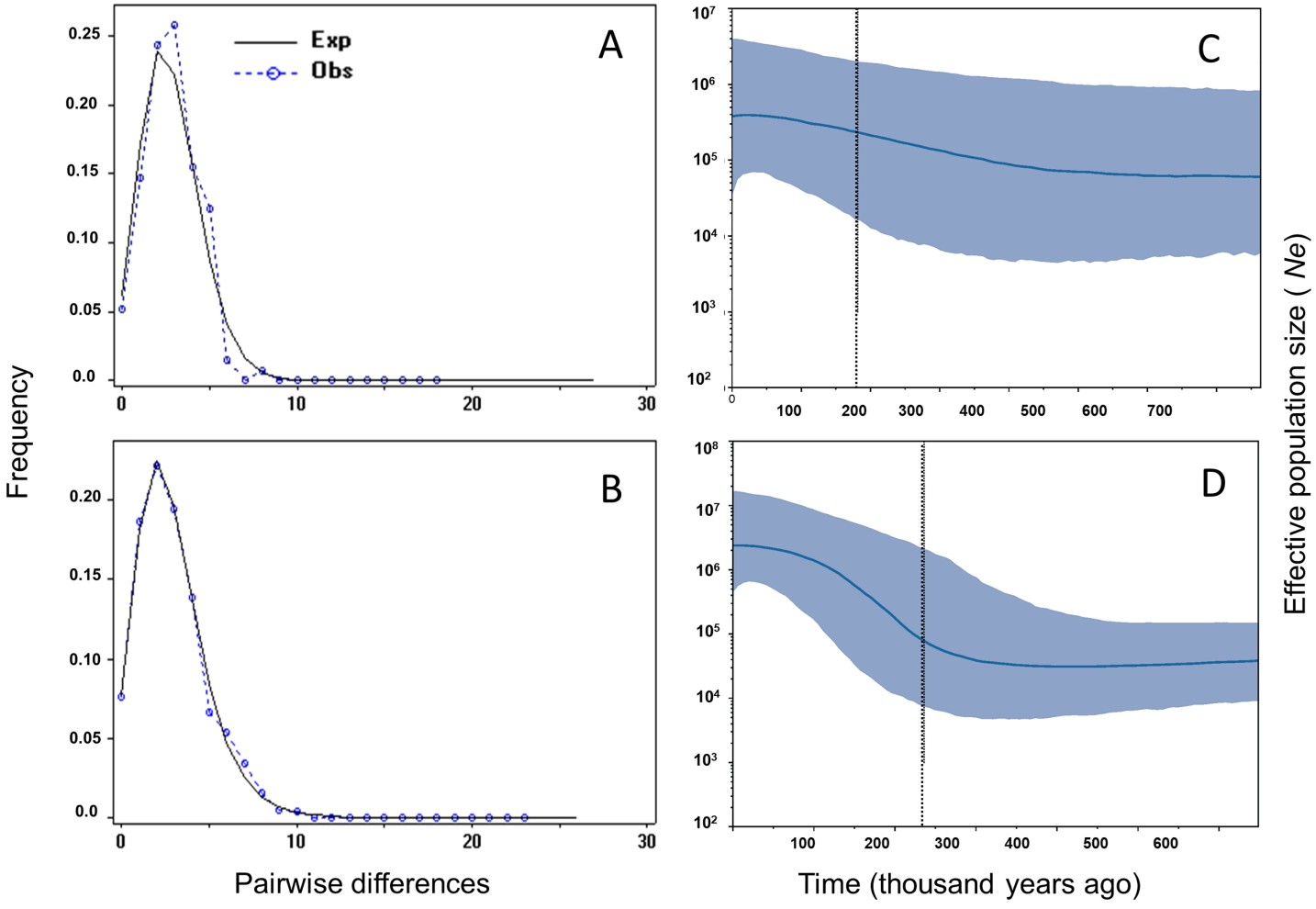

**Figure 3 Historical demographic analysis for *M. pedrazae* and *M. schiedeana*..** Mismatch distributions for a sudden expansion model in (A) *M. pedrazae* and (B) *M. schiedeana*. Bayesian skyline plots for (C) *M. pedrazae* and (D) *M. schiedeana*. The blue line traces the inferred median effective population size over time with 95% HPD shaded in blue. The black vertical line is the median projected on the expansion timeline.

years ago before the LGM, while for scenario 5, the admixture among the tree lineages was estimated around 22,900 (CI [1.7–72.7]) years ago, during the LGM. However, confidence intervals were large for both scenarios, going from the LIG until very recent times (Table S3 and Fig. S2).

## Present and past species distributions

The evaluation showed that the best niche model has a significant pROC ($P < 0.001$), 0.22 OR, and 1617.9 AICc with the LQP features and 1.0 RM. This niche model was constructed with the following variables and permutation importance: BIO14 (39%), BIO1 (34%), BIO3 (10.7%), BIO7 (6.2%), BIO18 (4.8%), BIO12 (2.7%), and BIO15 (2.7%).

The present niche model geographically projected showed the highest suitability values mainly at the center of Veracruz and for scattered areas towards the northern periphery, while Oaxaca showed moderate suitability values (Figs. 5A and 5B). The extrapolation

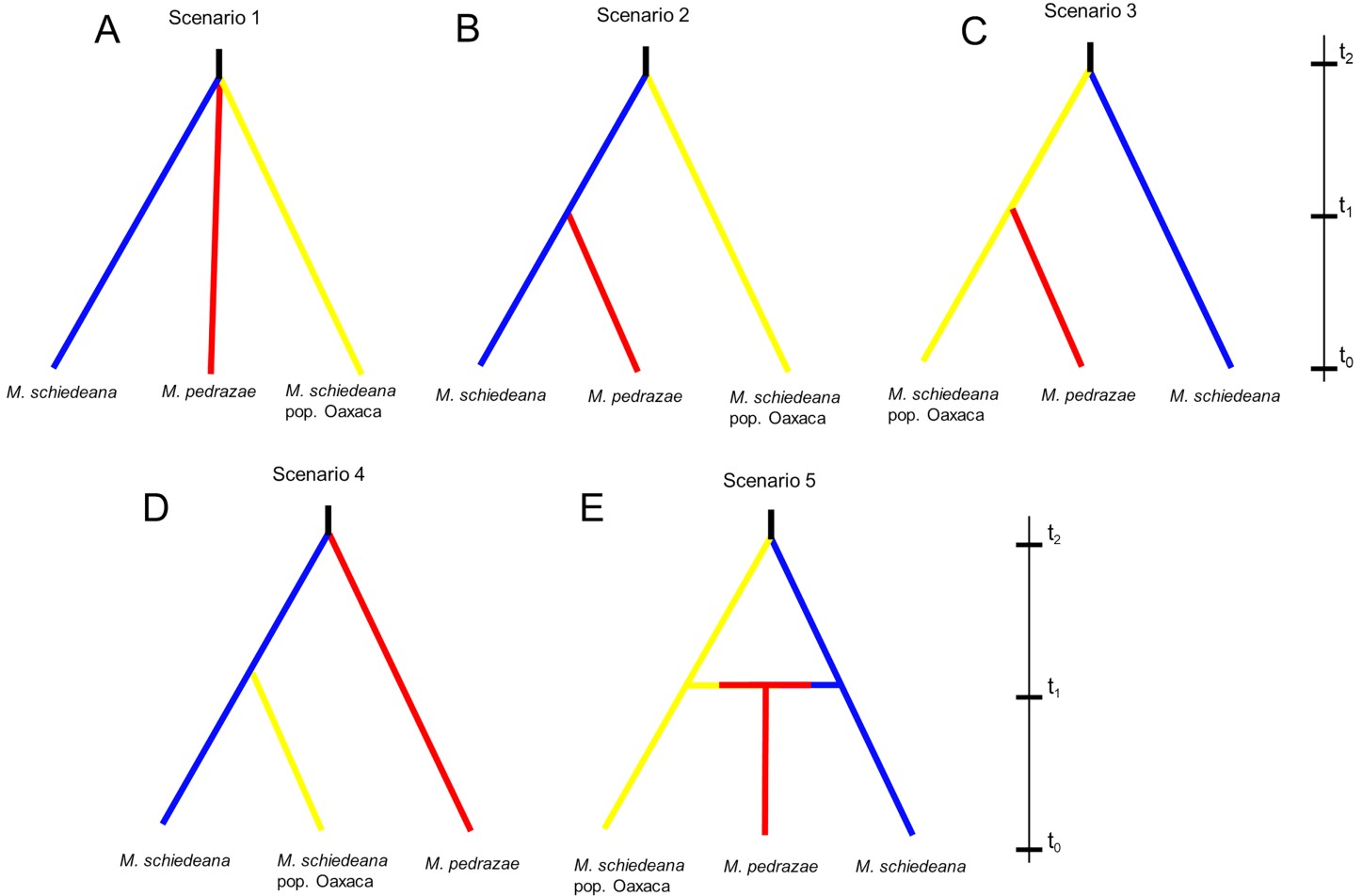

**Figure 4 Approximate Bayesian computation of five competing demographic scenarios for *M. pedrazae*, *M. schiedeana* and *M. schiedeana* pop. Oaxaca simulated in DIYABC.** (A) Scenario 1 predicts the three lineages diverging at the same time from a common ancestor (NA) at $t_2$ (stable model); (B) Scenario 2 predicts a split between *M. schiedeana* and *M. pedrazae* at time $t_1$, then both merged with *M. schiedeana* pop. Oaxaca at time $t_2$; (C) Scenario 3 predicts a split between *M. schiedeana* pop. Oaxaca and *M. pedrazae* at $t_1$ and subsequently both merged with *M. schiedeana* at $t_2$; (D) Scenario 4 predicts a split between *M. schiedeana* and *M. schiedeana* pop. Oaxaca at $t_1$ then both merged with *M. pedrazae* at $t_2$. These three scenarios correspond to a divergence model. (E) Scenario 5 predicts secondary contact between the three lineages at time $t_1$ (admixture model).

detection analyses showed that the percentage of non-analogous conditions for each past scenario were: NT1 (34.5%) and NT2 (3.24%) for MH, NT1 (4%) and NT2 (0.25%) for EH, NT1 (8.4%) and NT2 (0.13%) for LGM, and NT1 (24.9%) and NT2 (0.19%) for LIG. The NT1 and NT2 extreme values were low (−2.4 and −7.9) confirming low extrapolation risk (Fig. S3). On the other hand, the three methods of extrapolation for the four past scenarios showed similar predictions with only slight variations, therefore, only the projections with no extrapolation are shown.

Paleodistributions to the MH revealed that high suitability values were present in scarce areas at central east of the TMCF (Puebla and Veracruz) (Fig. 5C), whereas for the EH much higher suitability values were found across most of the species ranges from north (San Luis Potosí), central (Veracruz) and south (Oaxaca) (Fig. 5D). During the LGM high suitability habitat was present in two disconnected areas, one in central

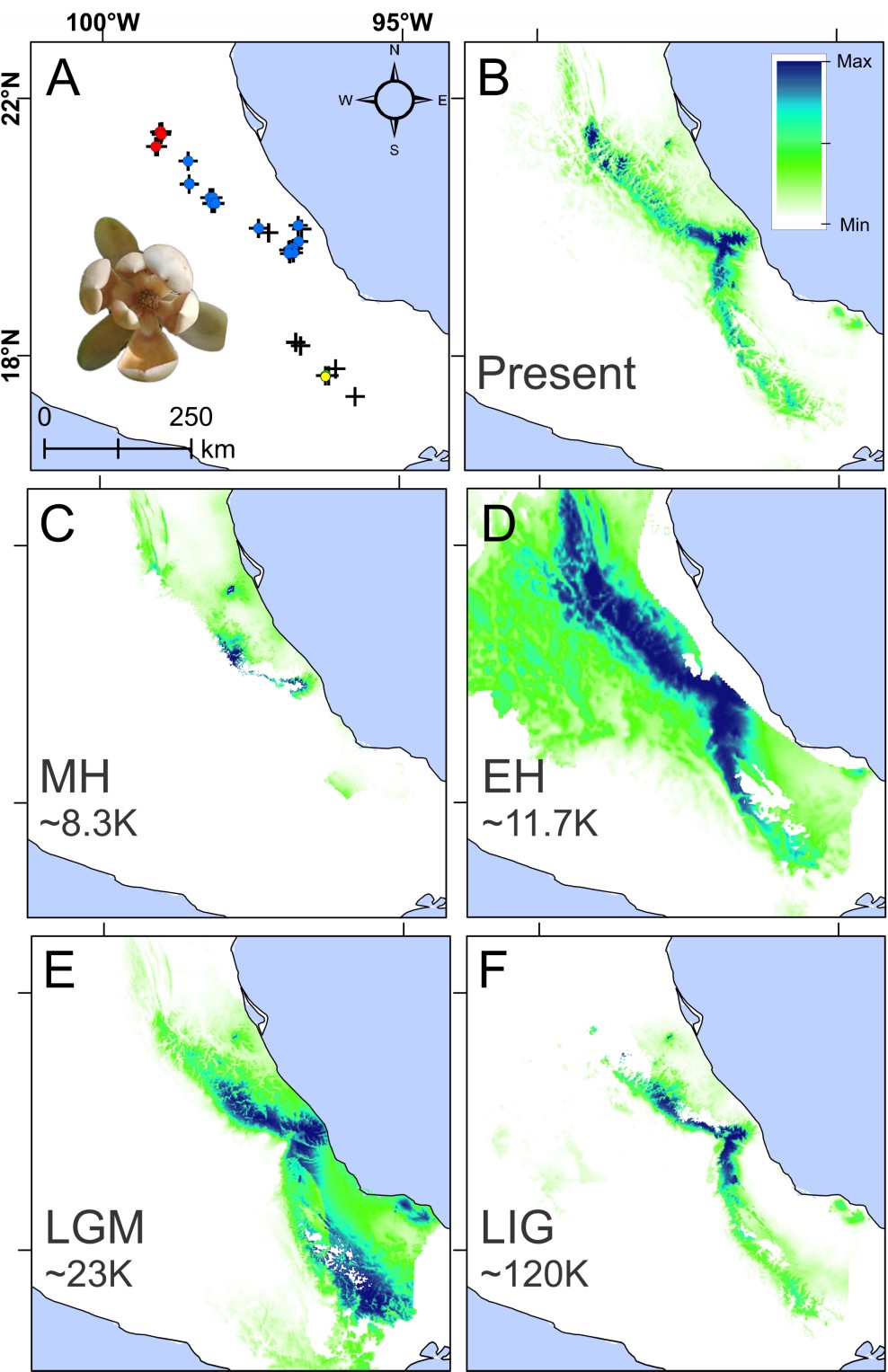

**Figure 5 Present and past species distributions modeling for *Magnolia pedrazae* (red dots), *M. schiedeana* (blue dots) and *M. schiedeana* pop. Oaxaca (yellow dots).** (A) The three lineage occurrences, (B) Present niche model, (C) Mid-Holocene (MH ~ 8.3 ka), (D) Early Holocene (EH ~ 11.7 ka), (E) Last Glacial Maximum (LGM ~ 23 ka), (F) Last Interglacial (LIG ~ 120 ka).

Veracruz and the other in south of Oaxaca (Fig. 5E). During the LIG, a thin strip with high suitability was restricted mostly at the central east (Veracruz and Puebla) (Fig. 5F). The Schoener's D and Pearson correlation between the present and past species projections were MH: D = 0.2 and r = 0.24, EH: D = 0.47 and r = 0.56, LGM: D = 0.48 and r = 0.53, and LIG: D = 0.49 and r = 0.69, which highlights the differences between the present and past niche suitability species distributions.

## DISCUSSION

In this study we employed analysis of phylogeography, historical demography and paleodistributions to understand how climatic oscillations of the Pleistocene have shaped geographical patterns of genetic diversity in endemic *Magnolia* trees of the largest remaining TMCF strip in Mexico. Based on four cpDNA sequences, we found evidence of three genetic lineages and signatures of demographic expansions, while past species distributions showed range expansions and contractions during the interglacial-glacial cycles of the Pleistocene, which suggest the strong role of climatic and/or environmental factors influencing the complex evolutionary dynamics of these *Magnolia* lineages.

### Genetic differentiation among TMCF regions

The MJ network, BAPS, and BPEC analyses jointly revealed three genetic clusters that showed a significant phylogeographical structure. The MJ network revealed 10 haplotypes exclusive to the north (*M. pedrazae*), 34 haplotypes to the central (*M. schiedeana*), and three to the south (*M. schiedeana* pop. Oaxaca) of the TMCF. Only two haplotypes were shared between the north and central lineages, one of which was widespread in the central region but not in the north. None of the haplotypes were shared between *M. schiedeana* pop. Oaxaca. Similarly, AMOVAs showed that the amount of genetic variance explained among these three lineages was moderate (48%), but it was considerable higher (58%) when only two lineages (*M. pedrazae* + *M. schiedeana* and *M. schiedeana* pop. Oaxaca) were considered. In contrast to these results, the Bayesian phylogenetic tree did not resolve the monophyly of the three lineages as all samples formed a polytomy with other closely related *Magnolia* species (*M. iltisiana* and *M. pacifica*). The lack of resolution of the phylogenetic relationships based on the cpDNA may be due to its slow-evolving mutation rate (*Richardson et al., 2013*) and incomplete lineage sorting, which has been found in other magnolias (*Kikuchi & Osone, 2021*). On the other hand, the chloroplast DNA in magnolias is maternally inherited (*Tobe, Abbott & Ballard, 1993*) and thus reflect patterns of seed-mediated gene flow. Bird dispersal as likely occurs for *M. schiedeana* (*Watanabe, Ikegami & Horie, 2002*) may provide gene flow at large spatial scales thus connecting populations over large distances (*Newton et al., 2007*; *Setsuko & Tomaru, 2009*).

The geographical circumscription of *M. schiedeana* has changed over time as the result of recent species designations based on morphological characters (*Jiménez-Ramírez et al., 2007*; *Cruz-Durán, Vega-Flores & Jiménez-Ramírez, 2008*; *Vázquez-García et al., 2012*; *Vázquez-García et al., 2013*), while molecular taxonomic studies are lacking. A population genetic study by *Rico & Becerril (2019)* pointed that the species designation

of *M. pedrazae* was not evident based on observed patterns of genetic differentiation estimated with nuclear microsatellites that showed strong to moderate differentiation across populations irrespective of their species identity. Although few populations of *M. schiedeana* were included in the study. For populations in Oaxaca, specifically in Santiago Comaltepec, the taxonomic designation remains unclear. Our results clearly support the genetic divergence of *M. schiedeana* pop. Oaxaca, which likely has remained isolated for a longer time according to the ABC analysis and the largest differentiation of this population shown by the pairwise $F_{ST}$ estimates. Due to our limited sampling for Oaxaca, we cannot confirm if this population represents a distinct taxonomic unit. A formal taxonomic delimitation study between *M. schiedeana* and recent designated species is outside the scope of this work. Species designation and delimitation remain controversial topics in biology (*De Queiroz, 2007*) and the taxonomy of Neotropical magnolias is a good example of the complexity to circumscribe species (*Vázquez-García, 1994*; *Aldaba-Núñez et al., 2021*). However, as recently suggested by *Huang (2020)*, to clarify whether population subdivision or speciation is warranted from an evolutionary perspective, we need a better understanding of the sources that generate and maintain structured intraspecific variation that may lead to interspecific divergence. In this study we contribute towards understanding the historical processes that have shaped pattens of genetic diversity in these *Magnolia* populations. Additional morphological, molecular (nuclear markers) and ecological data and broader population sampling is needed to further clarify species delimitation of the *Magnolia* section.

## Effects of the Pleistocene in the phylogeographical structure and demographic changes

Divergence among Neotropical magnolias is thought to occur mainly by vicariance given by the archipelago-type distribution of the TMCF (*Vázquez-García, 1994*; *Jiménez-Ramírez et al., 2007*; *Cruz-Durán, Vega-Flores & Jiménez-Ramírez, 2008*). Phylogeographical studies of tree species of the TMCF in Mexico have shown patterns of allopatric divergence caused by major biogeographical barriers, such as the Trans-Mexican Volcanic Belt (TMVB) (*Liquidambar styraciflua*, *Ruiz-Sánchez & Ornelas, 2014*) and for the Isthmus of Tehuantepec (*Palicourea padifolia*, *Gutiérrez-Rodríguez, Ornelas & Rodríguez-Gómez, 2011*) during the Pliocene to mid-Pleistocene. Specifically, the TMVB is a volcanic chain that horizontally splits Mexico into north and south, and which has profound impacts on the historical differentiation across multiple taxa, including flora and fauna (*Mastretta-Yanes et al., 2015*). The diversification of the *Magnolia* section has been estimated around 32.4 Mya at the early Oligocene, while the divergence of *M. schiedeana* can be placed at ~10 Mya during the late Miocene (*Dong et al., 2021*). This epoch was marked by the second pulse of volcanism from west to east of the TMVB (~11 to 7 Mya, *Ferrari et al., 2012*), and by warm and humid conditions (*Frigola, Prange & Schulz, 2018*). However, the estimated divergence based on the ABC analyses was far more recent; scenario 2 suggests that *M. schiedeana* pop. Oaxaca is the oldest lineage that diverged from a common ancestor around 80,000 years ago, while scenario 5 suggests that *M. schiedeana* and *M. schiedeana* pop. Oaxaca diverged from a common

ancestor around 85,000 years ago, both estimations of divergence events occurred during the warm and wet conditions of the LIG. The BPEC analyses suggested that ancestral haplotypes were in central Veracruz, and this coincides with the past niche projections where the largest extent of optimal habitat occurred in this region during the LIG. Our small sample size for *M. pedrazae* and *M. schiedeana* pop. Oaxaca, and the large confidence intervals obtained from demographic analyses, suggest that our divergence and demographic expansion dates should be interpreted cautiously. Moreover, the ABC simulations showed as equally likely two contrasting demographic scenarios, one that suggested the divergence of the three lineages, and the other their admixture, which we could not discriminate given their high type I and II error rates.

Neutrality tests showed demographic expansions for *M. schiedeana*, while the mismatch distributions and the star-shape topology of the haplotype network suggested that *M. pedrazae* also underwent a demographic expansion. These observations agreed with the Bayesian skyline plots that showed demographic expansions for *M. schiedeana* (~250,000 ya) and *M. pedrazae* (~200,000 ya), both predating the LIG. Demographic growth was also suggested by the ABC simulations, which showed an increase of population sizes for both scenarios after the divergence from a common ancestor around the LIG (Table S3). Our paleodistribution models during the last ~120,000 years ago highlighted complex population dynamics characterized by range shifts, contractions-expansions. Specifically, during the warmer conditions of the LIG, suitable habitat was mainly restricted to central Veracruz and Puebla, Then, during the LGM, suitable habitat remained in Veracruz and Puebla, but also was present in south of Oaxaca, although both areas were disconnected. Later during the early Holocene, suitable habitat considerably expanded northwards, from Oaxaca to Veracruz and San Luis Potosí. However, in the mid-Holocene suitable habitat was drastically reduced to small areas in Puebla and Veracruz, and then expanded in the present. Although our past scenarios were well supported and showed low extrapolation risk, caution must be warranted considering the low number of occurrences used to build the niche model and that these hypotheses are based on a single GCM, specifically in the LGM in which different GCM could give contrasting results (*Guevara, Morrone & León-Paniagua, 2018*). Increasing the number of occurrence records through field explorations and taxonomic studies in these taxa would further improve the accuracy of species distribution models.

The glacial refugia proposed by *Toledo (1982)*, suggests that Sierra de Juárez in Oaxaca and Córdoba in Veracruz were secondary refugia for tropical forest species during the LGM due to their high humid conditions. For instance, the Sierra de Juárez in Santiago Comaltepec is recognized as one of the TMCF regions where atmospheric humidity is the highest (*Gual-Díaz & González-Medrano, 2014*). The secondary refugia suggested by Toledo coincides with the locations of high suitable habitat predicted for the LGM, one towards the south of Oaxaca and the other in central Veracruz, although suitable habitat during the LGM increased relatively to the LIG, which the refugia model do not predict. As there is no conclusive evidence unambiguously supporting either of the two precipitation models, further investigations are needed by incorporating more samples (Oaxaca) and highly variable nuclear markers for testing these and other alternative

hypotheses (vulcanism in the TMVB), which could elucidate the complex evolutionary history of these lineages.

According to our genetic data, *Magnolia schiedeana* pop. Oaxaca likely remained isolated from the other two lineages before and during the LGM, as suggested by the lack of shared haplotypes among them and the evidence of its oldest divergence relative to the other two lineages. During the early Holocene (~11. 7 ya) the increase in temperature and humidity could have facilitated the range expansion from the south-central towards the northern periphery. Colonization of the northern periphery may have occurred by long-distance dispersal as suggested by the lack of IBD between *M. pedrazae* and *M. schiedeana*, and the retention of ancestral polymorphism in the cpDNA. The star-like network topology, the high haplotype diversity, but the low nucleotide diversity suggests the most recent recolonization of the northern periphery of the TMCF. The range contraction-expansion from the mid-Holocene (~8.3 ya) to the present suggested by the climatic models, may explain the weaker genetic divergence between *M. pedrazae* and *M. schiedeana* as there is no major geographical barriers that could explain their genetic divergence. Moreover, the BPEC analysis suggested that genetic divergence can also be shaped by environmental factors. Specifically, the environmental PCAs revealed significant differences among the three lineages, which mainly can be attributed to annual temperature range and precipitation seasonality. The TMCF of Mexico are composed of highly heterogenous forest that can vary markedly in environmental conditions (*Gual-Díaz & González-Medrano, 2014*). Local environmental differences have been suggested to generate the large intraspecific morphological variations observed across populations of *M. schiedeana* (*Vite, 2016*; *Rodríguez-Ramírez et al., 2020*). This environmental heterogeneity may have led to differential habitats preferences and thus to divergence by ecological factors, a potential hypothesis that could be tested in the future.

## Conservation implications

The TMCF is one of the most threatened ecosystems due to the strong anthropogenic pressures for land conversion and the predicted changes in temperature and precipitation regimes due to climate change (*Toledo-Aceves et al., 2011*; *Ponce-Reyes et al., 2012*). In Mexico, around 90% of *Magnolia* species are vulnerable to extinction by habitat loss and fragmentation, and these include *M. pedrazae* and *M. schiedeana* (*Rivers et al., 2016*). Genetic diversity estimates showed that San Luis Potosí is the most diverse region for *M. pedrazae*, and central Veracruz for *M. schiedeana*. However, only few of these studied localities (*e.g.*, CS and MA) are large populations (>300 individuals), while most of them are small (<100 individuals, *e.g.*, JH, FO, TU, MM, CP, MA). In fact, we found localities where the species might be locally extinct, such as TLA in Puebla and MI in Veracruz, as we found a single remaining individual. The TMCF of San Luis Potosí and Veracruz are under strong anthropogenic pressures from ongoing land-conversion to agriculture and livestock farming (*CONABIO, 2010*). Moreover, several populations do not occur under Protected Natural Areas, such as the most genetically diverse populations in

Veracruz. Protecting and restoring TMCF habitat is crucial for the conservation of *Magnolia* species (*Rivers et al., 2016*). This can be accomplished through implementing community-based management strategies, such as the promotion of sustainable and environmentally friendly tourism that can benefit the local communities. There are already successful examples of community-based programs in large remnant TMCF fragments of San Luis Potosí (La Trinidad) and Oaxaca (La Esperanza) that have large *Magnolia* populations (*e.g.*, FO, CS, SI, SC), and where local communities host ecotourist destinations and agreed the protection of their well-conserved forests forbidding commercial wood extraction.

On the other hand, our niche species distributions suggest that magnolias are sensitive to climatic events since historical times, which lead to range shifts, expansions, and contractions. Future climatic scenarios suggest a decrease of *M. schiedeana* suitable habitat by 2080 (*Vásquez-Morales et al., 2014*). Conserving genetic diversity of ancestral populations would be important for the species conservation as ancestral populations are sources of evolutionary adaptive potential (*Frankham, 2005*). According to our results, we suggest that special attention should be paid to populations in central Veracruz, as these are likely the most ancestral populations and where *M. schiedeana* is abundant in some localities; the same applies for the northern periphery in San Luis Potosí. Monitoring genetic diversity and evaluating the role of geographical and environmental factors on genetic diversity and gene flow would also be key for the *in-situ* species management in the long-term.

## ACKNOWLEDGEMENTS

We thank Bruno A. Gutiérrez Becerril, Sergio Nicasio Arzeta, and Benjamín Castillo Ponce for their assistance in collecting leaf material. We also thank the useful comments from three anonymous reviewers that improved the quality of our work during the revision stage.

### Funding

This work was supported by research funds to Yessica Rico by Instituto de Ecología A.C. Marisol A. Zurita-Solís received a CONACYT scholarship to pursue her master degree. The funders had no role in study design, data collection and analysis, decision to publish, or preparation of the manuscript.

### Grant Disclosures

The following grant information was disclosed by the authors:
CONACYT.

### Competing Interests

The authors declare that they have no competing interests.

## Author Contributions

- Yessica Rico conceived and designed the experiments, performed the experiments, analyzed the data, prepared figures and/or tables, authored or reviewed drafts of the paper, and approved the final draft.
- M. Ángel León-Tapia analyzed the data, prepared figures and/or tables, authored or reviewed drafts of the paper, and approved the final draft.
- Marisol Zurita-Solís performed the experiments, analyzed the data, authored or reviewed drafts of the paper, and approved the final draft.
- Flor Rodríguez-Gómez analyzed the data, authored or reviewed drafts of the paper, and approved the final draft.
- Suria Gisela Vásquez-Morales performed the experiments, authored or reviewed drafts of the paper, and approved the final draft.

## Field Study Permissions

The following information was supplied relating to field study approvals (*i.e.*, approving body and any reference numbers):

Permission to conduct our fieldwork was granted by the Mexican government Secretaria de Medio Ambiente y Recursos Naturales SEMARNAT permit # SGPA/DGGFS/712/1062/18.

## DNA Deposition

The following information was supplied regarding the deposition of DNA sequences:

The four chloroplast DNA sequences are available at NCBI GenBank: trnK5-matk; MW321790–MW321798, rpl32-trnL: MW321799–MW321808, trnS-trnG: MW321809–MW321812, trnT-trnL: MW321813–MW321827.

## Data Availability

The raw data are available in the Supplemental File.

## Supplemental Information

Supplemental information for this article can be found online at http://dx.doi.org/10.7717/peerj.12181#supplemental-information.

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
