# Peer review of "Influence of Pleistocene climatic oscillations on the phylogeography and demographic history of endemic vulnerable trees (section Magnolia) of the Tropical Montane Cloud Forest in Mexico"

_PeerJ, doi:10.7717/peerj.12181_

## Round 0.1 · original submission · Major Revisions

Dear Dr. Rico,

Overall the reviewers find the manuscript worthy of being published. Nevertheless, they found some major weaknesses that have to be addressed. In particular, the clarity of writing and argumentation has to be improved. Moreover, to strengthen the conclusions of the manuscript, the reviewers suggest some further analyses aimed to test your hypothesis. Please, respond point-to-point to the comments of reviewers to speed up the process of revision.

Once again, thank you for submitting your manuscript to PeerJ and we look forward to receiving your revision.

Sincerely,
Gabriele Casazza

Reviewer 1 ·

Basic reporting

Overall, the manuscript is clear but some section appear to be vague.
The methods section needs more clarity in reporting: bioclimatic variables, skyline plots.
The literature references are a bit insufficient and some references are missing to support the results.
The results are relevant to the hypothesis being tested, but the discussion shifts too much in the direction of species delimitation in the study system.

Experimental design

The research question is well defined.
Methods need more clarity and some choices need to be properly justified.

Validity of the findings

The sequence data have been deposited in GeneBank, but still not available. It is not clear if it would be possible to reconstruct the haplotypes with the sequences in GenBank.

Additional comments

I have now read the article "Phylogeography, lineage divergence, and paleodistributions support glacial refugia for an endemic vulnerable tree (Magnolia schiedeana) of the tropical montane cloud forest" twice, and I found it interesting and relevant in the context of cloud forest phylogeography in Mexico. Overall, the background is adequate and the methods are robust, but there are some issues here. However, I found myself reading the discussion and struggling between two main arguments: the phylogeographic history on the one hand, and the taxonomy of Magnolia schiedeana on the other. To me, too much focus is given to the latter and, according to the references cited, the present data are not the most adequate to test hypothesis around this issue. Regarding this point, authors appear to be going back and forth between accepting and not accepting the species separation. Finally, I found some flaws in the interpretation and discussion of the results, which might come down to a lack of clarity, but these weakens the relevance of the present work.

Below are more detailed concerns:
Lines 79-83: This is correct to a certain point. Yes, this is a (simplistic) expectation, but it has to be accompanied by a congruent time frame (see Ornelas et al. 2016). The dry refugia debate is centered around the last glacial period, and thus a demographic expansion pre-dating this period cannot be taken as evidence for this particular model (see the demographic expansion in Ramírez-Barahona & Eguiarte 2014). Also, Rogers & Harpeding (1996) should not be cited here.

Lines 94-96: The paper cited here for the age of the family (Azuma et al. 2011) –there is a typo in the year– does not have a divergence time analysis. Besides being unsupported, the 100 My age provided here seems a bit too old. The Angiosperm Phylogeny Website has crown Magnoliaceae to be younger than 100 Mya (this age is more in line with te stem age of the family); for instance, see the ages in Magallón et al. (2015) and Ramírez-Barahona et al. (2020), which appear also to be much younger. Please substantiate the North American origin.

Lines 165-168: the sequence data are not yet available, but if I'm understanding correctly only unique sequences were deposited, correct? My question is how would one reconstruct the haplotypes for individuals here? I'm not sure this would be possible with the information provided.

Lines 267: typo – substation rates

Lines 283-285. I'm having trouble with the testing data for the distribution models. Given the taxonomic uncertainty in the complex, these testing data are surely taxonomically messy as well. I find it inconsistent that authors only used 'occurrences from specimens collected in this study' for training 'due to taxonomic uncertainty' (typo in line 277 – uncertainly), but then choose to disregard this for the testing data. Please justify. Why not go with another strategy, 26 records are few, but still can be used to produce robust models; or why not extract data from other specimens vetted by the authors (or other experts).

Overall in the results and discussion, it is hard to follow all the acronyms for the states, I would suggest to spell them out in full. But, it is also hard to follow the discussion if not familiar with the geopolitical limits of Mexico. I would rather keep the discussion more broad in term of geography.

Lines 306-312. It is not clear which variables were downloaded. I'm assuming that for the LGM, the CHELSA variables were downloaded (based on the reported resolution). If so, I believe it is not ideal to use these in combination with the PMIP, because CHELSA and PMIP are based on different principles and algorithms. It can be done, but it is not ideal, so authors should be explicit about this choice. Also, I assume authors are using a single GCM on which the Paleoclim variables were constructed (if I recall correctly, this a version of CCSM). This has important consequences for downstream interpretations of the demographic history, so the choice should be properly justified.

Lines 369-375. These results should be backed up by a proper statistical test (e.g., discriminant analysis).

Lines 395-396. How was the timing of this sudden increase determined? This is a crucial point, because all the discussion hinges precisely on this timing. Going to figure 3, this definition of the ages for the sudden expansion is not clear.

Lines 444-446. It is clear that there is genetic structure, but this is far from being concordant with species delimitation.

Lines 461-465. Continued from the above comment. Here authors point the other way, that population genetic data does not support species delimitation. Also, this goes to my overall concern about the focus on the taxonomy of the species. I do not see how the present data is better to tackle this problem that the one used by Rico & Gutierrez-Becerril (2019) – this reference is missing from the reference list.

Lines 465-472. What is the relevance of all this? There are hints at some comparison being made, but the argument for doing so is not clear.

Lines 482-484. This is interesting. Does this mean, that the present data does not favor the segregation of M pedrazae?

Lines 487-493. This passage exemplifies the lack of clarity in the discussion. This section is vague and really speculative and without key information to back up the arguments. The TMVB might have restricted gene flow, but the time frame is critical here.

Lines 494-497. Authors are going back and forth between supporting the segregation of the two species and not supporting it.

Lines 517-525. Again the age cited appear to be too old (see Nie et al. 2008, Ramírez-Barahona et al. 2020), which would impact the proposed scenario. By the way, all this appear to be too vague and unsubstantiated. But the point here, is why is this history relevant? There is no connection between this passage and what comes next about population expansion during the last 300,000 years.

Lines 530-534. Which inter-glacial are you referring to here? Again, this concerns me because it is unclear how the estimated ages were determined. This is not clear from the skyline plots. Also, the statement on lines 533-534 does not hold with the evidence provided.

Lines 543-546. Ok, but see Ramírez-Barahona & Eguiarte (2014) to see that using different circulation models impacts the patterns seen in the species distribution models. So, the ensuing discussion is biased.

Lines 558-562. Sorry to be blunt, but this argument is plain wrong. As far as I'm concerned, the various species designations are not based on any population genetics. The only way that expansion-contraction dynamics can be linked to species designations would be to test that these species are in fact distinct genetic clusters.

Lines 566-587. These are a bit vague and authors should maker concrete proposals.

Figure 2. How was the ancestral haplotype defined (H23)? The colors here confused me because they do not correspond to the ones in figure 1. Are the colors related to inferred membership? The PCA plots need the % variance stated somewhere.

Figure 3. How where the location of the vertical lines defined in the skyline plots? These lines appear to mark the sudden expansion of the lineages, but this is not clear. Also the skylines show that after that period of time, demographic expansion continued, although authors state in the discussion that things were stable. Does M pedrazea merit analyses by its own?

Figure 4. The results presented here, to me, are not fully exploited in the discussion. Something else more complex appears to be happening here, which is not apparent in the skylines.

Reviewer 2 ·

Basic reporting

The authors study the Tropical Cloud Forest ecosystem and the implications of the climate oscillations of the Quaternary in the phylogeography and the historical demography of a vulnerable endemic tree, Magnolia schiedeana in Mexico. The authors apply genetic analysis in four regions of the chloroplast DNA and reconstruct the past distribution of the species with niche modelling. They aim to test the hypotheses of glacial refugia versus dry refugia. All in all, this work has good potential to be published with some further refinements in the analyses and the writing. In particular, the writing requires improvement for clarity as the text becomes ambiguous in parts and includes grammar or syntax errors. It would thus profit from the edits of a native speaker.
The background presented in the introduction requires improvement in logical flow and better link between the general research context and the specifics of the study area and species. Sometimes the text is hard for the reader to follow, e.g. the part referring to the dry refugia and the moist forests as theories explaining the dynamics of populations in response to climate oscillations. The latter is a key point that should be carefully addressed and in a more eloquent manner as it is tightly related to the hypotheses of the research presented so that the reader can better follow the research.
In the last paragraph of the introduction, the authors mostly present their technical aims rather than more general scientific questions. As such, they have not sufficiently linked the research objectives to a more general context in the sense of what new does this research bring beyond the knowledge regarding the study species - why is it interesting in a more general way.

Experimental design

Most of the methods used are in general sufficiently described but there is room for improvement particularly in improving the link between the genetic data and the ecological analysis. It would also be interesting if the authors could show the niche overlap of the projections in % over time in the last paragraph of the results (e.g. with ecospat package) and test with statistical tests whether the number of pixels has significantly changed. Please see the uploaded word file for some more specific comments.

Validity of the findings

Please see the uploaded word file for some more specific comments.

Annotated reviews are not available for download in order to protect the identity of reviewers who chose to remain anonymous.

Reviewer 3 ·

Basic reporting

In this paper the authors assessed the phylogeography, historical demography and Quaternary distributions for an endemic and endangered species of the Tropical Montane Cloud Forest (TMCF), Magnolia schiediana. The study is based in four chloroplast DNA regions and past species distribution models. They found phylogeographic structure corresponding to three lineages: west, central-east and south TMCF of Mexico. Genetic data and past SDMs suggest a demographic expansion from Veracruz that occurred in the Pleistocene (~200,000 – 300,000 years ago). According to the authors, their results support the Pleistocene refugia hypothesis for species from the Tropical Forest in Mexico.

The manuscript is written in clear unambiguous, professional English, with a few details that can be easily addressed (see general comments). For readers that might not be as familiar with the geography of Mexico, I suggest using the complete name of the Mexican states throughout the text instead of abbreviations.
Literature and references are sufficient and according to the field background, and enough context is provided.

The manuscript is well structured. The figures and tables are clear, but I suggest eliminating Table 1 from the main manuscript, because small sample size may represent a caveat for population comparisons and conclusions at the population level. Also, I recommend adding a figure of the Bayesian phylogeny in the main text, but if authors consider that this figure won't add to the study, at least include it as supplementary material (see general comments).

The aims of the study are stated in the introduction. The analyses are adequate and results are very interesting as presented by the authors. Analyses are in accordance to the objectives, but I have some concerns regarding the conclusions provided by the authors and suggest performing a formal hypothesis test by means of an ABC framework (see validity of the findings section). In addition, given the small sample size for some localities the scale of the analyses should be reconsidered (see comments below).

Sequences have been deposited to GeneBank and accession numbers are provided in the manuscript. Moreover, sequence alignment is included as supplementary material.

Experimental design

This research is within the scope of the journal, the research question is well-defined. The authors clearly state the knowledge gap filled by this study. Also, the authors provide a clear statement of the objective of the study. Nevertheless, re-phrasing the aims in terms of the hypotheses tested by their analysis may provide a wider public; i.e. “In this study we (i) tested the phylogenetic relationship between M. pedrazae, M. schiediana and M. schiediana var. Oaxaca; and (ii) we used a phylogeographic approach and past species distribution models to characterize the historical demography and changes in distribution ranges of these taxa". I would add (iii) using an ABC framework to test the dry-refuge vs. moist-forest hypothesis (see section 3).

Finally, one of the objectives was to identified populations that should be priorities for conservation based on their levels of genetic diversity. Nevertheless, a small sample size for some localities may bias these conclusions; thus, providing conservation guidelines at the population level should be avoided, conservation guidelines could and should be provided but at a wider regional level.

Methods are mostly well explained and conform to technical standard, but a few details should be added. In particular I’m concerned about the small sample size for some populations (i.e. one locality in Veracruz and one in Puebla has n=1, M. pedrazae var Oaxaca has n=5). This has important implications when estimating levels of genetic diversity and genetic structure at population levels and, therefore, all analyses and result should be conducted only at the lineage level. This should also be considered for implications for conservation.

Validity of the findings

The benefit to literature is clearly stated, all underlying data have been provided and the analyses are robust.

Nevertheless, some conclusions should be revised, and alternative interpretation of the results should be considered. In this sense, the data presented by the authors may be providing better support to the moist-forest hypothesis:

Past species distribution models shown in figure 4 suggest that range was smaller during the LIG (not the LGM), because this is the time when we can see the smaller suitable area for the presence of the species. Then the models suggest that for the LGM the species expanded towards the south reaching Oaxaca (possible secondary contact with the lineage found in Oaxaca?), and remained there through the Early Holocene (EH); then from the EH to the Mid-Holocene (MH) a displacement of the species distribution into the north reaching SLP (possible secondary contact with M. pedraza?). These models support downslope migration and expansion promoting population connectivity through the LGM according to the moist-forest hypothesis as stated in Ramirez-Barahona and Eguiarte (2013). According to Fig. 4 we can see a slight range contraction from the MH into the present species range, which could relate to upslope migration and range fragmentation in accordance to the moist-forest hypothesis (Ramírez-Barahona and Eguiarte, 2013). These results taken together with genetic data suggest that lineages pre-dating the LGM would be present (reported divergence time for M. schiediana is 11 mya) and show marked differentiation, but these lineages would have an overall wide distribution (no IBD when Oaxaca is removed from the analyses further supporting a moist-forest hypothesis) (Ramírez-Barahona and Eguiarte, 2013). Genetic diversity would be favored (high overall genetic diversity, with at least four haplotypes with high frequency [H03, H02, H36 and H12]) and lack of genetic structure (one genetic cluster with wide distribution for M. schiedeana according to BAPS, with an area of higher genetic diversity in TLA and MI, and may be even GO, which agrees with areas of environmental stability in the distribution models), and signals of demographic expansion before the LGM (> 200,000 ya), followed by range expansion after the LIG, from the LGM to the Holocene (high number of low frequency haplotypes connected by few mutational steps in the haplotype network).

Of course, not all signals completely agree with the moist-forest hypothesis, because under this hypothesis we wouldn’t expect signals of demographic expansion (Ramírez-Barahona and Eguiarte, 2013), and neutrality tests, mismatch distribution and skyline plots all suggest demographic expansion for M. schiediana.

Therefore, and considering that the authors have very good data, I strongly suggest conducting a formal hypothesis testing based on an ABC framework (Cornuet et al., 2008, 2010, 2014; Knowles 2009; Bertorelle et al., 2010) that would assign a probability to each scenario and make an explicit test for the dry-refuge vs. moist-forest hypotheses and facilitate scenario choice. This will also allow testing if current levels of genetic variation and genetic structure in M. schiediana could be explained by other geographic or geologic events that do not relate to Pleistocene climate change.

Additional comments

Line 9-10. Please mention in the abstract the number of populations and individuals included in the analysis.

Line 50. Please consider changing phrasing to: “; which implies that it is highly vulnerable to climate change”

Line 58. Please consider changing this phrase to “These processes led to complex…” instead of “These processes had led…”.

Line 63-66. Please review the text “The dry refugia states that cooler conditions during the Last Glacial Maximum (LGM ~23-14 ka) shifted species migrations to the lowlands but the prevalence of arid conditions led to species displacements and contractions into glacial refugia; subsequently population expansions and long-distance recolonizations took place with the increase in temperature and humid conditions in the Holocene…)”. Please make the dry refugia hypothesis more explicit, because I got a little confused when reading the text.

Lines 120- 122. Please mention the total number of populations or localities included in the analysis.

Lines 136-143. Is there any information regarding mechanisms for seed and pollen dispersal? Are there any reports about the dispersal distance?

Line 145. In the Sampling and DNA sequencing section, please provide the total number of samples per taxon used in the analyses. How many M. pedrazae, M. schiedeana and M. schiedeana var Oaxaca individuals were analyzed?

Lines 152-153. Did the authors consider a distance between sampled individuals to avoid or minimize chances of sampling only closely related individuals?

Line 203. Some of the localities listed in Table 1 show a very low number of samples (n=1), therefore, genetic diversity is not well sampled for these locations and comparisons and conclusions at the population level should be avoided. I suggest moving this table to supplementary materials and focusing the paper and its conservation implications only at the lineage level.

Line 225-227. Please include original references to the methods employed by BEPC: Manolopoulou et al. 2011 and Manolopoulou and Emerson 2012. Also, please provide the underlying assumptions for this test. This is particularly important because the test assumes that there is phylogeographic structure and that current population structure is explained by founder events followed by migration (isolation by distance model) (see Manolopolou et al. 2020). The latter implies that the data supports the Pleistocene refuge hypothesis; therefore, there is a bias regarding the dry-refuge vs. moist-forest models. Perhaps, the authors should test first which of the two models or hypotheses has the best support given their data and depending on that implement this test.

Line 235. Please make it explicit that the program BEPC estimates the best number of migration events.

Lines 240-241. I recommend performing Mantel tests before BEPC, to support that data is consistent with an isolation by distance model and justify the use of BEPC analysis.

Lines 277-291. Please make the final number of occurrence points for each taxon used for the analysis explicit and clarify whether data for the three taxa or only for M. schiediana was used to build the models.

Lines 306-313. Please explain how the AOGCM to be used in the analyses was defined or provide an argument to justify the use of a particular AOGCM.

Line 320. Please explain where the 106 individuals came from; is it because of the outgroup? In methods section, the authors mention a data matrix of 96 individuals including some sequences downloaded from the GeneBank.

Lines 323-324. Please provide the results from MrBayes analysis (phylogenetic tree) as a figure in the main manuscript.

Lines 345-348. As previously mentioned (Line 203), some of the localities listed in Table 1 have only one sample. Genetic diversity at the population level should be taken with caution since these measures are particularly sensitive to sample size, and comparisons and conclusions at the population level should be avoided. I suggest moving table 1 to supplementary materials (or totally removing it from the manuscript) and focusing the paper only at comparing and assessing genetic diversity at the lineage level.

Lines 358-363. Instead of explaining the genetic clustering obtained with BAPS from a population perspective, the authors should emphasize that the three genetic clusters correspond to regions of the TMCF, which can be classified into south, central and north, and also is consistent with the classification of three taxa: M. pedrazae, M. schiediana and M. schiediana var Oaxaca.

Lines 363-369. As I mentioned above, I have some concerns regarding the BPEC analysis because of the assumptions underlying the analysis. In this case in particular, data suggest the presence of three lineages, which could have diverged in a different site each; therefore, we cannot rule out the possibility of more than one ancestral location. I recommend avoiding making inferences regarding ancestral haplotypes or ancestral locations, at least at this point.

Line 370-375. I recommend conducting a statistical analysis, such as MANOVA, to test for statistical differences in the environmental envelope occupied by each lineage (Nakazato et al., 2010 AJB 97(4):680-693; Di Febbraro et al. 2017 J Biogeog 44:2828-2838).

Lines 376-377. As mentioned above, sample size for some populations is very low and I recommend taking analyses at the population level with extreme caution and, if possible, avoiding making conclusions and inferences at this level. Please keep only pairwise FST between lineages or taxa.

Lines 378-383. It is very interesting that M. schiediana var Oaxaca show the highest level of genetic differentiation and that IBD analysis losses power when removing this data. Nevertheless, according to table 1 only 5 individuals from one locality from Oaxaca were included in the analysis and the geographic distance of this locality to the other taxa seems considerable (Fig 1). Does this provide enough evidence in itself to infer that this group or lineage is well-supported or might this be an effect of having samples from only one site in Oaxaca?

Line 396. Please add ya after ~250,000.

Lines 392-396. For the Bayesian Skyline Plots, each lineage may show a distinct independent historical demography, as seems to be the case for the data shown by the authors. I recommend keeping M. schiediana and M. pedraza separate for the historical demography analyses, because joining the data may produce a bias if the signal of demographic expansion for one of the lineages is very strong. It is interesting that signal of population expansion occurred about the same period (but temporality should be interpreted with caution; see comment on Lines 530-534) for both taxa, further suggesting they have constituted a different lineage for a long time perhaps even before the Pleistocene (moreoever since authors mentioned that divergence for M. schiedeana was estimated 11 Mya).

Lines 415-420. It is not clear if the model and past projections show distribution changes for the three taxa or only for M. schiediana. Please, clarify this in the methods section and in the legend for figure 4. Also in figure 4 legend define the meaning of MH, EH, LGM and LIG.

Lines 481-485. Data suggests that M. schiediana var. Oaxaca could have been isolated for a longer time. We should take these results with caution because we cannot rule out that sampling could be causing some bias since only one locality (N=5) from Oaxaca (and with a long distance from the other sampling sites; fig 1A.) was included in the analysis. I see that authors mention the need to gather additional data in lines 511-512, but suggest this caveat of the data should be considered since the beginning of the discussion.

Line 517. Please clarify “high mid-latitudes”. The term is ambiguous, so I recommend providing a latitude interval or threshold.

Line 521. Does “icehouse conditions” refer to glacial conditions?

Line 525-527. I recommend providing the date of divergence of M schiedeana in the introduction or in the study species sections. This will provide context to the results of the past species distribution models and historical demography data.

Lines 530-534. Results from Bayesian Skyline plots should be taken with caution, because (1) small sample size at least for M. pedrazae and M. schiediana var Oaxaca, which reduces the power of the analysis; and (2) transforming x-axis to time based on a standard substitution rate estimated for angiosperm chloroplast. Rates of molecular evolution vary considerably (see Smith and Donoghue, 2008 Science 322(5898):86-89; Gaut et al. 2011 Annu Rev Ecol Evol Syst 42:245-266; Lanfear et al., 2013 Nature Communications 4:1879), and this has an effect on the estimated dates. Therefore, the discussion should focus on consistent signals of demographic response across lineages instead of focusing on the date per se.

Line 533-542. Please, for a better context, provide dates of divergence for the three lineages included in the study (if available from the literature), because if the divergence pre-date the climate changes of the Pleistocene this would have implications for the interpretation of the results. The results from past species distribution show interesting shifts in species distribution during the distinct periods and may even support hypotheses of secondary contact between lineages during these times or the possibility of incomplete lineage sorting or divergence associate to Pleistocene climate change. Nevertheless, if M. schiedeana diverged 11mya, it is possible that M. pedraza and M. schiedeana var Oaxaca diverged before the Pleistocene. Estimates of the dates of divergence between lineages are needed for the reader to better-understand the processes that have affected these taxa.

Line 543. Please, change to “According to the glacial refugia hypothesis, …”.

Lines 543-562. Is it possible to have multiple refugia instead of just one? Please, perform a sum of the models to have a better visual representation of the areas of environmental stability that might have been refugia.

Lines 570-577. Given the small sample size for some localities, the genetic diversity of some sites might be underrepresented. Therefore, providing conservation guidelines at the local level may not be adequate, and conservation implications should be mention only on a broader regional level.

---

## Round 0.2 · Major Revisions

Dear Dr. Rico,
Overall the reviewers find the manuscript strongly improved.
Nevertheless, they suggest that some methodological choices the use of climatic data from two different datasets and the use of different resolutions like must be more clearly justified. If the extent of the region is enough to use rasters at 2.5 min of resolution without losing information or changing the inferences, why do you also use data at 30sec of resolution? Why do not use 2.5 min for all the analyses?

Please, respond point-to-point to the comments of reviewers to speed up the process of revision.

Once again, thank you for submitting your manuscript to PeerJ and we look forward to receiving your revision.

Sincerely,
Gabriele Casazza

Reviewer 1 ·

Basic reporting

no further comments

Experimental design

no further comments

Validity of the findings

The interpretation of the results and conclusions are more cautious and some limitations of the data are identified.
Still, I have an issue with the circulation model used for the SDMs.

Additional comments

This is the second time reviewing this paper, and I congratulate the authors for being positive to my previous comments. The authors have done a great job revising their paper, which in my opinion is now a stronger paper with relevant results. These data would be a great addition to the growing evidence on cloud forest phylogeography in the region.
Some comments:
1) Lines 360-362. This is a erroneous interpretation of the results presented in Guevara (2020), who shows that different circulation models give contrasting results. I'm would recommend authors add a sentence or two in the discussion stating that the results and interpretation of SDMs are based on a single GCM, and thus caution is warranted. Or something like along these lines.
2) Lines 599. change 'do' for 'does'.
3) Figure 2. Authors should consider changing the colors in the figure, to make these less primary, and to keep in mind that it might be difficult to discern between colors if color-blind.
4) Figure 4. Same as above for the colors. Also, it would be nice to highlight the scenario with the greatest support (perhaps a square around the scenario).

Reviewer 2 ·

Basic reporting

The authors have overall improved their manuscript. Yet, some of the comments from the previous round of review have been partly addressed and particularly some of the methodological issues. Nevertheless, the concept and topic addressed are relevant, contribute to the field, and merit publication.

Experimental design

The methods used to address the questions are relevant. However, the niche analysis has room for improvements as already suggested in the previous round of review.

Validity of the findings

The findings are highly recommended to be refined in the light of improvements in the analysis of the projected distributions.

Additional comments

Please find below specific comments that could improve this work.

L. 325: How did you define this radius? Is it e.g. the average density of occurrences?
L. 330: All occurrences or only the ones from GBIF?
L. 340: Based on the paper you cited in your response:

"collinearity shift and environmental novelty can negatively affect Maxent model transferability. We therefore recommend to quantify and report collinearity shift and environmental novelty to better infer model accuracy when models are spatially and/or temporally transferred."

This is one more reason to keep a lower threshold for collinearity in variables.

L. 341: A threshold of 0.85 is still too high. You could at least lower it at 0.75 e.g. see Elith et al., 2012. This will also help you reduce the number of variables which is way too high for so few occurrences. Therefore, there is high risk of model overfit and low transferability in time and space.

L. 358-368: Comparisons of niche and space overlap under different resolutions over space and time cannot be objective as the resolution can drammatically alter predictions of SDM. Please see Randin CF, Engler R, Normand S et al. (2009) Climate change and plant distribution: local models predict high-elevation persistence. Global Change Biology, 15,1557–1569.

For Mid Holocene there is a dataset from worldclim paleodata at 30s resolution you could use. See here http://www.worldclim.com/paleo-climate1. For other climate layers you could downscale the data.

L.369-371: For what % of the study area the variable values are beyond the calibration range (novel climate)? You could test this before projecting and select for the variables with the best transferability for more objective projections of the models.

L477: The ratio of occurrences should have been something around 70% training and 30% testing. Not the other way round...For such a small number of occurrences, this number of variables is excessive. A rule of thumb is one variable per 20 occurrences... See Guisan et al 2017, Habitat suitability models in R (book)

L. 479: Did you let the algorithm decide the type of features? It would be easier for a model to extrapolate the simpler it is e.g. you could exclude hinge features at least (if not threshold too) and leave Q, L, P, T.

L.653: The last paragraph is very interesting as it gives a clear hands-on recommendation.

Minor comments:
L.126: Please correct to "considerable"
L. 247: Please correct to "Manolopoulou" in the second citation

Reviewer 3 ·

Basic reporting

Authors made a good job addressing my previous comments. There are still a few issues and typos, see General comments for the author.

Experimental design

Overall genetic and spatial analyses are adequate, and previous comments have been addressed.

Authors followed my suggestion of running an ABC framework to explicitly test the dry-refugia and moist-forest hypotheses. Nevertheless, more detail is needed in the method section so analyses can be replicated. Moreover, the authors mention that the five tested scenarios were “considered as the most plausible given the observed relationships of divergence and admixture among the three lineages”. Regarding the original question of testing two biogeographic hypotheses, I suggest considering at least two divergence scenarios (or group of scenarios), one that considers divergence associated with Pleistocene climate changes (t2 and t1), as already done; and another that considers divergence associated to vulcanism in the Transmexican Volcanic Belt (t4 and t3) (see for example, Pérez-Crespo et al. 2017 J Biogeog 44:2501-2514; Ramírez-Barahona and Eguiarte 2014 J Biogeog 41:2396-2407).

In addition, in the results section authors mentioned that two scenarios (2 and 5) obtained equal probabilities, and type 1 and type 2 error rates were high. To select between these scenarios, authors performed two additional runs in DIYABC where each scenario was alternatively removed from the analysis. This is not an adequate method to select between these competing scenarios and a different approach should be used (see Beaumont et al. 2002 Genetics 162:2025-2035; Beaumont 2010 Annu Rev Ecol Evol Syst 41:397-406; Bertorelle et al. 2010 Mol Ecol 19(13):2609-2625; Robert et al. 2011 PNAS 108:15112-15117). In this sense, from the methods section it is not clear if previous runs were conducted to test different combinations of summary statistics and parameter values to adjust scenarios. The selection of summary statistics is important, because it will add or subtract power to the analysis. In addition, by conducting previous runs we can identify if tested scenarios are sufficiently different so the analysis will be able to distinguish between them or identify if some of the parameters should have wider or narrower distributions. Conducting previous runs to adjust summary statistics and parameter distribution and adding scenarios with deeper dates of divergence will probably solve this issue.

Validity of the findings

Previous comments have been addressed but see comments on experimental design regarding the ABC analyses.

Additional comments

Line 25. Correct italics in the s of populations.

Line 117. Please change “at the north of Oaxaca” to “in northern Oaxaca”.

Line 260-263. Please add a references that support the use of MANOVA and Tukey tests to assess environmental differentiation.

Lines 291-303. It is important to provide detailed information on the summary statistics, paramater values and parameter distribution used when conducting coalescent simulation such as DIYABC, because the quality of the results will depend on the quality of initial data (see Bertorelle et al. 2010. Molecular Ecology 19(13):2609-2625 https://doi.org/10.1111/j.1365-294X.2010.04690.x). Moreover, authors should provide all information so the analysis can be replicated. Therefore, methods used to run DIYABC need a more detailed explanation. Please, define the distribution model used for each parameter (uniform, logistic, etc.) in addition to the median, maximum and minimum values for each parameter (N1, N2, N3, NA, t1, t2, ra) defined initially to run DIYABC. Particularly as time is an important parameter being tested, please define the mean values and confidence intervals defined for t1 and t2 in both generetion time and its equivalent in geological time.

Line 323. Please change “spatial accuracy sufficient to produce” to “spatial accuracy to produce” or to “they have sufficient quality and accuracy to produce”

Lines 322-330. Please explain why using a thinning of 3 km for one dataset and 5 km for the other.

Line 333. Please provide a justification for using CHELSEA.

Line 342. Please review this sentence “and the Pearson’s correlation threshold of 0.85 using NTBOX v0.1.4.5 R package (Osorio-Olvera et al., 2020) was performed…” It is a little bit odd.

Lines 348-351. Also review sentence “Some levels of…, resulted in 100 candidate niche models” Perhaps change to “Some levels of…, which resulted in 100 candidate niche models”.

Line 362-368. Selection of climate input data for species distribution modelling could influence the results (Bobrowski and Schickhoff, 2017). Therefore, please provide a justification for using data from WorldClim in some cases and data from CHELSA in other cases to obtain the species distribution models and to perform past projections.

Line 417. Regarding results from AMOVA, where you say that “… a slightly lower proportion of the genetic variance and genetic differentiation were observed relative to the other two groupings” Do you mean relative to the two lineage grouping? because I see in table 2 and in the results that FCT value for the three lineage grouping is higher than FST for no grouping but lower than FCT for three lineages grouping.

Lines 432-434. That haplotypes H10, H17 H22, H23, H36 and H40 were assigned to M. schiedeana pop. Oaxaca is not clear from Fig. 2. Only one haplotype from population Mi is colored yellow in panel 2D. In the haplotype network (panel 2B) haplotypes H43,H44 and H45 are assigned to the M. schiedeana pop. Oaxaca cluster.

Lines 435. Supplemental S2 shows pairwise FST values. I’m not sure why it is cited here.

Lines 453-456. Bayesian skyline plot show a steeper expansion for M. schiedeana than for M. pedrazae. Is the result for M. pedrazae signficant?

Lines 463-467. For DIYABC results, the logistic regression shown in supplementary materials, shows equal probabilities for scenario 2 and scenario 5, as well as high type 1 and type 2 error rates. Therefore, the correct implementation should be treating both scenarios as equally possible, and results should highlight both of them. Of course, running DIYABC while eliminating scenario 5 will result in scenario 2 having the best probability, and the opposite if scenario 2 is excluded, because this type of analysis will only contrast and assign a probability to the scenarios that are fed to the program. We should remember that we are only selecting the best scenario among those that are being tested and, we should not treat these scenarios as the absolute truth because we cannot rule out that there are other untested scenarios that might be equally plausible or even have a better fit to the observed data. Therefore, the proposed method to discrimate between scenarios 2 and 5 do not seems adequate. Previous runs should be conducted to adjust parameter values and to assess if selected summary statistics have enough power to differentiate between competing scenarios, this will result in higher reliability for scenario choice; and if even then the analysis is unable to discrimate between scenarios 2 and 5, then both shall be presented as equally plausible (please read Beaumont et al. 2002 Genetics 162:2025-2035; Beaumont 2010 Annu Rev Ecol Evol Syst 41:397-406; Bertorelle et al. 2010 Mol Ecol 19(13):2609-2625; Robert et al. 2011 PNAS 108:15112-15117).

Line 524. Change “due to is slow-evolving” to “due to its slow-evolving”.

Lines 571-574. It is difficult to assess if estimated dates of divergence are reliable, because of lack of detail regarding the description of methods and some methodological issues regarding scenario choice. Moreover, apparently the main differences between the tested scenarios related to identifying the ancestral lineage but not to differences in times of divergence. In this sense, since Magnolia section diversified ~10 Mya and signals of population expansion in M. schiedeana were dated ~250,000 ya, it would be interesting if competing scenarios also included differences in times of divergence, and test for divergence driven by Pleistocene climate changes (dates in accordance to t2 = LIG and t1 = LGM) vs. divergence driven by vulcanism pulses of the TMVB (t4 and t3).
Lines 581-583. Please make it explicit in the methods section that tested scenarios also considered changes in effective population sizes (varNe).

Line 588. Please change prolong to extend

Line 598. According to line 303, the five tested scenarios were considered as the most plausible given the observed relationships of divergence and admixture among the three lineages. Nevertheless, as I understand it, the ABC approach was implemented to explicitly test the dry-refugia and moist-forest hypotheses; thus, scenarios should include differences in times of divergence and changes in effective population sizes (for example, Pérez-Crespo et al. 2017 J Biogeog 44:2501-2514; Ramírez-Barahona and Eguiarte 2014 J Biogeog 41:2396-2407). If these changes are already considered in the five analysed scenarios, this should be explained in the methods section.

---

## Round 0.3 · Minor Revisions

Dear Dr. Rico,

the reviewers find the manuscript strongly improved. However, there is still a main issue about the number of variables used in the model, the proportion of occurrences used for validation and some minor issues that have to be solved before acceptance. Please, respond point-to-point to the comments of reviewers to speed up the process of revision.

Once again, thank you for submitting your manuscript to PeerJ and we look forward to receiving your revision.

Sincerely,
Gabriele Casazza

Reviewer 2 ·

Basic reporting

The authors have greatly improved their manuscript in many aspects and it reads much better than previous rounds and I would like to congratulate them for all the work they put. However, there is a couple of points regarding the niche modelling analysis that have been partially addressed.

Experimental design

The authors have improved their niche modelling approach by downscaling some of the paleoclimate data where needed and by producing less complex models which facilitate the transferability of models in time. Also,they tested the degree of the models extrapolation over time which showed interesting results, that could be a bit more emphasised in their results and discussion sections. Figure S3 is very nice. In this regard,they could test the degree of extrapolation of the used variables to see which aspects of the climate differ the most over the different climate periods. Also, they could mention in the supporting information how they did the downscaling (e.g., calculated the climate anomalies and interpolated in the study area; see Patsiou et al., 2014 Global Change Biology). However, there are a few flaws in how they select the cadidate variables:
a) The number of variables is still too high for the number of occurrences, despite they lowered the correlation coefficient threshold. A good rule of thumb for the number of variables is about 1 variable per 20 occurrences. Therefore, I would recommend that they include the test they mention on the variable values of their occurrence data, but also they carry out the same test for all variables in the whole study area. Then, based on the consensus of the two approaches select the least variables possible. Additionally, they could consider the known ecological preferences of the species and further exclude variables if needed.
b) Lines 502-507 could go to the methods. However, why are the authors using a higher proportion of the occurrence data for validation of the models than for the model training? That should have been the other way round as has been mentioned in the previous round of review. Please adjust your analysis accordingly.

Validity of the findings

The final results could be improved if the authors address the points mentioned in field 2 and further improve their niche modelling analysis. If they succeed to do so, the article could eventually be published.

Reviewer 3 ·

Basic reporting

The authors have made a good job addressing my previous concerns. Nevertheless, I still have some comments that should be addressed carefully. Please see General comments to the authors (in particular the last four comments: from lines 483 forward).

Experimental design

no comment

Validity of the findings

no comment

Additional comments

Line 310 Three million datasets under each scenario for 5 scenarios is 15 million simulations. It this correct? Please clarify

Line 319 The meaning of “these uniform a priori can be considered non-informative” is not clear. Please clarify sentence. In additio, change a priori to priors.

Line 313-314 To avoid possible confusions please state 100 - 100 000 generations considering a 10-year generation time.

Line 314 Please change adding up to added.

Lines 315-316 Please change sentence to “For the final run, we used the following summary statistics… It is not necessary to repeat the parameters again, since their values were not modified to adjust the models.

Lines 317 and 318 Please change "segregation sites" to "segregating sites".

Line 319 Please change from "chloroplast-wide" to "choroplast".

Line 320 Please move up the information about generation time as suggested in comment for lines 313-314.

Fig S2. Please also provide the figure for the PCA (model check) for scenario 5.

Lines 483 - 488 Please remove these lines since, as I mentioned in my previous review this type of test is not valid to discriminate among scenario 2 and scenario 5. Regarding this, please acknowledge in the discussion that high type I and type II error rates make it difficult to discriminate among two scenarios and high CI for parameters suggest that scenarios for the divergence of Magnolia species might be complex and tested scenarios might not be depicting such complexity.

Lines 601 - 604 Please state that these results should be taken with caution, as the difficulty to differentiate among two scenarios and the oldest date of divergence included in the analysis was 1,000,000 years ago, you cannot rule out an oldest age of divergence and these results suggests that M. schiediana has a complex evolutionary history. Moreover, your Skyline plot analyses suggest that population expansion occurred around 250,000 and 200,000 years ago, which is older than the ~80,000 years for divergence estimated by DIYABC, providing further support to the argument of a complex evolutionary history and that the tested scenarions might not be covering the complete picture.

Lines 626-630 should be moved up (between lines 608-609), before starting the discussion about the results from the neutrality tests.

Lines 630-631. I suggest eliminating the phrase “Despite of these uncertainties, by integrating our whole results, we lean towards the dry-refugia model”. In my opinion, this sentence is inadequate as results do not provide enough power to lean towards the dry refugia hypothesis or leaning towards the moist-forest hypothesis, particularly because hypothesis testing through coalescence in DIYABC couldn’t resolve between the admixed and non-admixed models. The data presented in this manuscript show that there were changes in the distribution and demography of Magnolia schiedeana associated to Pleistocene climate change. Nevertheless, lineage divergence may pre-date the Pleistocene epoch and Pleistocene climate change might have only promoted expansion and contraction and probably some secondary contact, which may explain incomplete lineage sorting found in the genealogy estimated with Beast. Moreover, I consider that the authors conclusion at the end of the discussion that “However, as there is no conclusive evidence unambiguously supporting either of the two precipitation models, further investigations are needed by incorporating more samples (Oaxaca) and highly variable markers for testing these and other alternative hypotheses (vulcanism in the TMVB) driving lineage divergence” is adequate and cautious enough.

---

## Round 0.4 · accepted · Accept

Dear Dr. Rico,

Considering that in the new version you have pointed out the limitations of your analysis suggested by the reviewers, I am very pleased to inform you that your paper " Influence of Pleistocene climatic oscillations on the phylogeography and demographic history of endemic vulnerable trees (section Magnolia) of the Tropical Montane Cloud Forest in Mexico " is accepted for publication in the PeerJ. Congratulations!

Thank you for submitting your work to PeerJ.

Sincerely,
Gabriele Casazza